# LongCoT: Benchmarking Long-Horizon Chain-of-Thought Reasoning

**Sumeet Ramesh Motwani** [*1]  **Daniel Nichols** [*2]  **Charles London** [*1]  **Peggy Li** [*2]  **Fabio Pizzati** [*3]
**Acer Blake** [1]  **Hasan Hammoud** [4]  **Tavish McDonald** [2]  **Akshat Naik** [1]  **Alesia Ivanova** [1]
**Vignesh Baskaran** [5]  **Ivan Laptev** [3]  **Ruben Glatt** [2]  **Tal Ben-Nun** [2]  **Philip Torr** [1]
**Ameya Prabhu** [6†]  **Brian Bartoldson** [2†]  **Bhavya Kailkhura** [2†]  **Christian Schroeder de Witt** [1†]

Correspondence: `charles.london@cs.ox.ac.uk, sumeet.motwani@eng.ox.ac.uk`

[Website]    [Benchmark]    [Code]

## Abstract

As language models are increasingly deployed for complex autonomous tasks, their ability to reason accurately over longer horizons becomes critical. An essential component of this ability is planning and managing a long, complex chain-of-thought (CoT). We introduce LongCoT, a scalable benchmark of 2,500 expert-designed problems spanning chemistry, mathematics, computer science, chess, and logic to isolate and directly measure the long-horizon CoT reasoning capabilities of frontier models. Problems consist of a short input with a verifiable answer; solving them requires navigating a graph of interdependent steps that span tens to hundreds of thousands of reasoning tokens. Each local step is individually tractable for frontier models, so failures reflect long-horizon reasoning limitations. At release, the best models achieve <10% accuracy (GPT 5.2: 9.8%; Gemini 3 Pro: 6.1%) on LongCoT, revealing a substantial gap in current capabilities. Overall, LongCoT provides a rigorous measure of long-horizon reasoning, tracking the ability of frontier models to reason reliably over extended periods.

## 1. Introduction

The ability of language models to reason reliably over long chains of thought becomes critical as they are deployed on

---
* Joint-first authors. † Equal advising. [1]University of Oxford [2]Lawrence Livermore National Laboratory (LLNL) [3]MBZUAI [4]KAUST [5]Hexo AI [6]University of Tübingen. Correspondence to: Charles London <charles.london@cs.ox.ac.uk>, Sumeet Ramesh Motwani <sumeet.motwani@eng.ox.ac.uk>.

*Proceedings of the 43$^{rd}$ International Conference on Machine Learning*, Seoul, South Korea. PMLR 306, 2026. Copyright 2026 by the author(s).

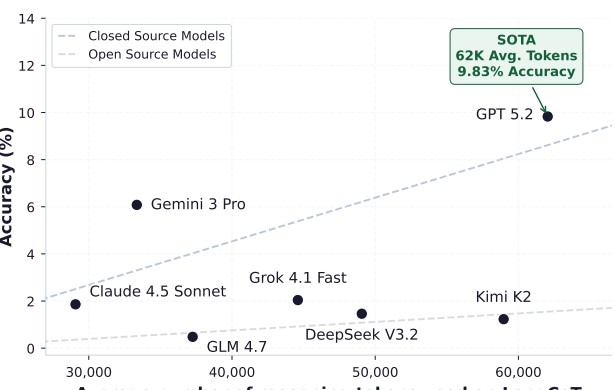

*Figure 1.* Accuracy versus token usage on LongCoT. GPT 5.2 achieves 9.83% with an average of 62K output tokens per problem.

increasingly harder tasks. Current benchmarks measure this capability only indirectly, either through hard but short reasoning problems or through agentic workflows where tool use and scaffolds leverage the underlying models' reasoning abilities. As context limits grow and test-time scaling is adopted widely, an important frontier emerges: the fundamental ability of models to maintain a coherent reasoning trace over long horizons. In this work, we design a benchmark to measure and study this capability in order to ask:

> *How accurately can frontier models reason over long horizons in their chain of thought?*

We define long-horizon reasoning as the ability to reason reliably over many interdependent steps in an extended chain of thought. Complex tasks require models to (i) plan, explore, and backtrack; (ii) maintain long-term state by prioritizing relevant context; (iii) monitor progress and discover errors or dead-ends; and (iv) perform credit assignment by linking errors or progress to specific prior steps. Current benchmarks rarely stress-test these abilities. FrontierMath (Glazer et al., 2024) caps generations at 10K tokens. HLE (Phan et al., 2025) typically induces fewer than 5K reason-

## Simplified Long-Horizon Reasoning Problems (3 of 5 LongCoT Domains)

### 🧪 Chemistry

**Reaction Cascade** — 9 subproblems with **explicit** ($\rightarrow$) dependencies

*# Each step produces/selects a molecule; later reactions combine earlier outputs.*

$N_1$: Identify molecule from structural properties $\rightarrow$ Mol-1.

$N_2$: Match structure string to known molecule $\rightarrow$ Mol-2.

$N_3$: Predict product of Mol-1 + Mol-2 $\rightarrow$ Mol-3. — combines $N_1, N_2$

[... *identify Mol-4 via formula; select Mol-6, Mol-7 from candidates...*]

$N_5$: Predict product of Mol-3 + Mol-4 $\rightarrow$ Mol-5. — combines $N_3, N_4$

$N_8$: Predict product of Mol-6 + Mol-7 $\rightarrow$ Mol-8. — combines $N_6, N_7$

$N_9$: Compute molecular diameters for Mol-5, Mol-7, Mol-8. — combines $N_5, N_7, N_8$

***Answer:*** Diameters $= [11, 5, 5]$

### 🔢 Mathematics

**Chained Competition Problem** — 14 subproblems with **explicit** ($\rightarrow$) dependencies

*# 14 competition problems chained: each answer parameterizes the next.*

$N_1$: Consecutive integers with product divisible by $k$.

   backtrack: $k$ depends on $N_2$ and $N_{11-14}$, creating a cyclic dependency.

$N_2$: Nested-squares geometry; branches into two chains. — uses $N_1$

$N_3$: Find a volume $\rightarrow N_4$: Evaluate a sum. — from $N_2$

[... $N_5 \rightarrow N_6 \rightarrow N_7$: *further competition problems...*]

$N_8$: Find roots $\rightarrow N_9$: Count divisors $\rightarrow N_{10}$: Count reflections. — $N_9 \leftarrow N_4$; $N_{10} \leftarrow N_8$

$N_{11}$: Optimize $\rightarrow N_{12}$: Primes $\rightarrow N_{13}$: Angles $\rightarrow N_{14}$. — from $N_2$; $N_{14} \leftarrow N_{12}$

***Answer:*** $(N_{10}, N_4, N_1, N_{13}) = (9, 5, 5, 27)$

### ♟ Chess

**Minimax Pawn Capture** — Problem has **implicit** subproblems with dependencies

*# The dependency graph is a game tree that emerges from the rules.*

Setup: $30 \times 30$ board with a knight at $(7, 13)$ and 8 pawns at given positions.

Game: Two players alternate turns choosing which pawn the knight captures next (moving via shortest path). Alice must maximize total moves; Bob minimizes. Find the total moves under optimal play.

Search: 8! orderings naively; prunable via minimax with memoization.

***Answer:*** 106 moves

## Computational Structure

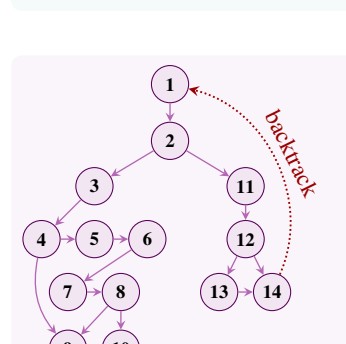

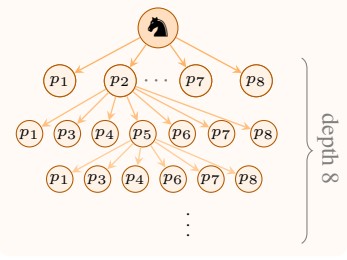

*Figure 2.* **LongCoT problems demand long-horizon reasoning.** Each of our five domains requires constructing and traversing a computational dependency graph in a long chain-of-thought. These graphs can be DAGs, search trees, cyclic graphs, constraint graphs, or execution traces. Frontier models struggle with LongCoT problems: even the best model **(GPT 5.2) achieves only 9.83% accuracy.**

ing tokens on average. Agentic benchmarks evaluate complex multi-step workflows, but domain-specific tool use and scaffolding dominate improvements (Merrill et al., 2026). Long-context benchmarks (Bai et al., 2024; Magic, 2024) test retrieval over long inputs but require only short outputs. What remains an open problem is directly measuring the inherent capabilities of models to keep reasoning as their chain of thought grows longer.

To this end, we introduce LongCoT, a benchmark of 2,500 problems from chemistry, mathematics, computer science, chess, and logic. Each problem consists of a short prompt (median 2K tokens) with a verifiable final answer; solving it requires navigating a graph of interdependent subproblems that span tens to hundreds of thousands of reasoning tokens.

Constructing problems that require such extended reasoning is non-trivial. Our domain experts design several domain-dependent parameterized templates that are either *explicit* compositions (where the prompt exposes a directed graph of meaningfully composed subproblems), or *implicit* (where a single question requires planning and search through a large graph structure). This allows us to produce meaningful problems for different domains that are atypical in public corpora and enables scalable question generation (Zeng et al., 2025) by simply modifying parameters. We also control for single-step difficulty: atomic steps or subproblems are tractable in isolation, and failures on LongCoT isolate limitations in long-horizon capabilities. Specifically, the difficulty primarily arises from the dependencies and the need for planning,

*Table 1.* **Comprehensive benchmark comparison.** LongCoT uniquely combines short inputs (<6K tokens), long reasoning outputs (10K–100K+ tokens), controlled single-step difficulty, no tool dependencies, and verifiable answers across five diverse domains. Existing benchmarks either test short reasoning chains, require long inputs, depend on tools, or do not control for step-level difficulty.

| Benchmark | Input Context | Long-Horizon Reasoning Needed | Scalable Difficulty | Reasoning Isolation | Task Domain | Best Score |
|---|---|---|---|---|---|---|
| ***Frontier Reasoning*** | | | | | | |
| FrontierMath (Glazer et al., 2024) | Short | ⊖ (≤10K tok) | ✗ | ✓(HC) | Math | 31.3% |
| HLE (Phan et al., 2025) | Short | ⊖ (≤8K tok) | ✗ | ✓(HC) | Multi | 38.3% |
| FrontierScience (Wang et al., 2025) | Short | Unknown | ✗ | ✓(HC) | Multi | 25.2% |
| ***Long-Context Benchmarks*** | | | | | | |
| LongBench v2 (Bai et al., 2024) | Long | ✗ | ✓ | ✗ | Multi | >65% |
| MRCR v2 (Vodrahalli et al., 2024) | Long | ✗ | ✓ | ✗ | Synthetic | >95% |
| ***Agentic Benchmarks*** | | | | | | |
| SWEBench-Verified (Jimenez et al., 2023) | Short | ✓ | ✗ | ✗(SC) | Code | 80.9% |
| WebArena (Zhou et al., 2023) | Short | ⊖ | ✗ | ✗(SC) | Functional | 71.6% |
| BrowseComp (Wei et al., 2025) | Short | ⊖ | ✗ | ✗(SC) | Web | 77.9% |
| Gaia2 (Froger et al., 2025) | Short | ✓ | ✗ | ✗(SC) | Multi | 42.1% |
| PaperBench (Starace et al., 2025) | Long | ✓ | ✗ | ✗(HC) | AI Papers | >40% |
| TerminalBench (Merrill et al., 2026) | Short | ✓ | ✗ | ✗(HC&SC) | Code | 64.9% |
| HCAST (Rein et al., 2025) | Short | ✓ | ✗ | ✗(SC) | Code | N/A |
| **LongCoT (Ours)** | **Short** | ✓ | ✓ | ✓ | **Multi** | **9.83%** |

**Reasoning Isolation:** HC = hardness confounded as success also requires esoteric knowledge; SC = scaffold confounded as success also requires tool use.

error detection, managing context, and backtracking.

LongCoT is intended to test long-horizon reasoning as a fundamental model capability. Some of our domains are algorithmically structured, and this is precisely why they are useful: they provide clean, scalable, and exactly verifiable tests of the abilities we wish to measure. Allowing code execution would let models offload the dependency structure externally rather than carry it through their chain of thought. Our primary evaluation therefore tests the model alone, with no tools or scaffolding. This isolates the abilities we wish to measure: planning, maintaining state over large dependency structures, propagating constraints, and backtracking. We separately evaluate a scaffolded track with code execution in Section 4.2 and find that while it improves performance on some procedural domains with programmatic solutions, our compositional problems (Mathematics, Chemistry, Computer Science) are more challenging.

At release, frontier models achieve less than 10% accuracy on LongCoT. Notably, GPT 5.2 achieves 9.83% with 62k tokens on average per question, significantly higher than other frontier models tested. The gap between performance on existing benchmarks and ours suggests models struggle to reason reliably over long outputs, where context degrades, plans drift, partial results are lost, models give up early, and errors go undetected (Arike et al., 2025; Sinha et al., 2025; Zhou et al., 2025). In Section 4.2, we analyze how performance varies across models, problem types, and horizon lengths, along with common errors. These results expose a gap in model capabilities to perform long-horizon rea-

soning that is central to automating complex, economically valuable tasks. Our contributions can be summarized as:

- We introduce LongCoT, a hard benchmark of 2,500 problems spanning chemistry, mathematics, computer science, chess, and logic, designed to directly measure long-horizon reasoning as a model capability.

- We develop a scalable method to construct problems requiring extended reasoning through expert-designed parameterized templates, where difficulty arises from the graph-structured sequence of meaningfully composed steps or subproblems rather than forcing models to perform arbitrary operations repeatedly.

- We evaluate several open and closed source frontier models and find that even the best achieve less than 10% accuracy, revealing a substantial gap between performance on existing reasoning benchmarks and reliable long-horizon reasoning. We provide detailed analysis of failure modes and how performance varies across models, domains, and horizon lengths. Furthermore, we separate out LongCoT-mini, which comprises easier problems that allow us to better differentiate the long-horizon performance of open-source models.

## 2. Related Work

We provide an overview of related benchmarks in Table 1 and compare them to ours using several criteria. We broadly divide the literature into benchmarks focused on

hard reasoning, long-context (input), and agentic tasks.

**Frontier Reasoning Benchmarks.** Benchmarks like MATH (Hendrycks et al., 2021), AIME, FrontierMath (Glazer et al., 2024), and HLE (Phan et al., 2025) test mathematical and scientific reasoning at the frontier. While they can include challenging and esoteric questions, these benchmarks do not typically induce completion lengths greater than 10K tokens, as Table 1 illustrates. In particular, these benchmarks do not isolate and test long-horizon reasoning behaviors critical to performance at 100K+ reasoning tokens, including the ability to develop plans, track increasingly many variables, maintain reasoning chain coherence, and suppress error accumulation. LongCoT focuses on testing these long-horizon reasoning behaviors by using (implicitly or explicitly) composed problems, where each step is challenging but tractable and performing well requires solving all steps correctly within a single very long chain of thought. We follow a template based design that enables programmatic difficulty scaling and an analysis of performance degradation by scaling problem lengths. This ensures LongCoT remains challenging as model capabilities improve.

**Long-Context Benchmarks.** Long-context evaluations are a well-established component of benchmarking. However, such prior work typically either investigates very high token count model inputs (e.g. prompting a model to find a needle-in-a-haystack), or relatively low token count model outputs. For e.g. LongBench v2 (Bai et al., 2024) and MRCR v2 (Vodrahalli et al., 2024) evaluate on very large-scale model inputs with multi-step reasoning-based retrieval tasks, while PlanBench (Valmeekam et al., 2023), LongProc (Ye et al., 2025) and LongGenBench (Liu et al., 2024) evaluate models that generate moderately sized outputs limited at 8K tokens. LongCoT is, to our knowledge, the first evaluation to test long-output performance by demanding very long chains of thought (often containing 100K+ tokens) that steadily progress through a series of individually tractable subproblems connected in a graph-like structure (see Figure 2).

**Agentic Benchmarks.** Agent benchmarks like Terminal-Bench (Merrill et al., 2026), PaperBench (Starace et al., 2025), and HCAST (Rein et al., 2025) test models with problems requiring multi-step reasoning chains and tool use. These evaluations measure an important capability: whether a full system can succeed in realistic environments. However, because benchmark performance depends jointly on reasoning ability and agent proficiency (tool use, domain-specific scaffolds, etc.), they do not isolate the underlying model's ability to generate effective, very long horizon chains of thought. Notably, agents often fail due to reasoning errors compounding (Zhu et al., 2025), failures to change early plans (Xinmiao et al., 2026), and degradation over task horizons due to model capability limitations (Back-

lund & Petersson, 2025; Luo et al., 2025). By separating reasoning from tool orchestration, LongCoT can directly extract signal about long-horizon reasoning effectiveness.

## 3. Benchmark Construction

### 3.1. Structure

Each problem $x$ in LongCoT is a short self-contained prompt (median 2K tokens, max 6.7K) with a domain-specific verifier $V_x : \mathcal{Y} \to \{0, 1\}$ that checks a final answer $y \in \mathcal{Y}$. The model receives $x$, produces an extended chain of thought (often exceeding 50K tokens), and outputs $y$, which is scored by $V_x(y)$. We focus on *long-horizon* problems, where the correct final answer depends on getting many dependent intermediate steps correct, so that models must sustain correct multi-step structure over long reasoning traces. We distinguish instances by whether the dependency structure relevant to solving is *explicitly presented* in the prompt or only *implicitly specified* by rules/constraints.

**Explicit (compositional) templates.** A compositional template $T = (V, E, \{g_i\}, \phi)$ specifies an explicit dependency DAG. The nodes $V = \{1, \dots, n\}$ index subproblems, each with a unique correct answer $a_i$. An edge $(i, j) \in E$ means that solving the subproblem at node $j$ requires the correct answer $a_i$ from node $i$. For each node $j \in V$, the instantiation function

$$g_j(\lambda_j, \{a_i : (i, j) \in E\}) \mapsto (q_j, a_j)$$

produces a concrete subproblem $q_j$ together with its correct answer $a_j$, given template parameters $\lambda_j$ and the answers of its parents. Errors in earlier nodes cam propagate to all downstream subproblems. The prompt is the concatenation of these composed subproblems $\{q_i\}$, and the final answer is $y = \phi(\{a_i\}_{i \in V})$. Long-horizon difficulty is controlled by the depth/width of the explicit DAG and by constructions that induce branching. We use four motifs when composing subproblems (see Figure 3): (i) *linear chains* with sequential dependencies; (ii) *DAGs* with multiple parents per node; (iii) *conditionals* where intermediate solutions determine which branches to reason over; and (iv) *forced backtracking* where a node specifies a target output and the model must search over candidate inputs to a non-invertible function.

**Implicit (procedural) templates.** Procedural templates specify the dependency structure *implicitly* via rules or constraints, rather than providing it explicitly as a finite subproblem DAG. We write an implicit template as $T = (\mathcal{S}, s_0, R, Q)$, where $\mathcal{S}$ is a configuration space, $s_0$ is the initial configuration, and $R$ is the rule/constraint specification (e.g. a transition relation for planning/simulation, or a set of constraints for CSPs). The rules $R$ induce a latent structure over $\mathcal{S}$, such as: a state-transition graph (planning); a constraint/factor graph (e.g. Sudoku); or a game tree with

| Math | | Chemistry | | Computer Science | | Logic | | Chess | |
|---|---|---|---|---|---|---|---|---|---|
| Algebra & Functions | 25% | Mol. Properties | 33% | Code Tracing | 25% | Planning / Search | 30% | Simulation | 29% |
| Number Theory | 24% | Reaction Pred. | 22% | Scheduling | 20% | Dynamic Prog. | 20% | Best Move | 21% |
| Geometry | 20% | Mol. Topology | 13% | Graph Algs. | 18% | Counting | 20% | Minimax | 21% |
| Combinatorics | 20% | Substructures | 13% | Type Inference | 15% | Pathfinding | 10% | Pathfinding | 15% |
| Probability | 8% | Mol. Representations | 10% | Distributed Sys. | 12% | Optimisation | 10% | Placement | 7% |
| Sequences & Series | 3 % | Ring Analysis | 9% | Compiler Passes | 10% | CSP | 10% | Retrograde | 7% |

EXPLICIT (COMPOSITIONAL TEMPLATE EXAMPLES)     IMPLICIT (PROCEDURAL TEMPLATE EXAMPLES)

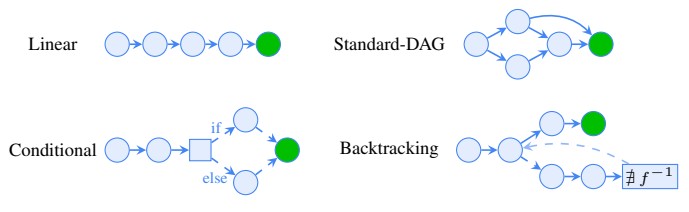
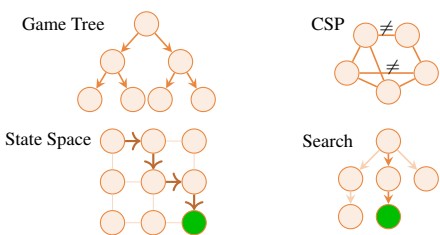

*Figure 3.* **LongCoT problem domains and reasoning structures.** *(Top)* Distribution of subtopics across five domains. *(Bottom)* Example dependency graphs. **Explicit** templates present the graph directly in the prompt; **Implicit** templates require models to discover and navigate latent structure (game trees, constraint satisfaction, simulation). Actual problem graphs are significantly larger and more diverse than these schematic examples. This diversity across domains, graphs, and question types is essential for testing long-horizon reasoning: one problem might not isolate all relevant capabilities (managing context, backtracking, planning, execution), but the full benchmark does.

adversarial branching (e.g. chess). The required output is specified by a template-level *query Q* on this structure. These include reaching a goal state, producing a satisfying completion, or minimising an objective.

Both classes of templates require reasoning over several interdependent steps. Compositional templates make dependencies between subproblems or steps explicit, while search templates implicitly encode these. Combined, this allows us to build useful domain-specific questions that test the core capabilities required for long-horizon reasoning, including:

- *Context management and recall* require tracking and correctly retrieving information that matters many steps later. Both template categories involve aggregating states from much earlier in the reasoning chain.

- *Self-evaluation and backtracking* involves detecting errors and returning to the last correct state. Errors such as constraint violations or incorrect intermediate values may only become apparent late in the reasoning chain.

- *Planning* capabilities are required to organize subtasks and solve problems efficiently. Search templates do not expose the solution structure, so models must plan ahead to minimize backtracking and redundant exploration.

- *Exploration and execution* is needed to evaluate candidate branches and sustain correct reasoning over extended CoT. Procedural templates require efficiently considering multiple paths; compositional templates require reliably executing many dependent subproblems.

A central design principle of LongCoT is that problems are

evaluated without external tools, as this would change the capability being measured. Some of our domains admit efficient programmatic solutions, and that is precisely why they are useful: they expose long dependency structures in clean, controllable, and exactly verifiable form. Real-world tasks are typically messier, less formally specified, and harder to decompose correctly. In such settings, a system cannot reliably offload the reasoning burden without first identifying the right state, decomposition, and invariants to maintain across a long trajectory. LongCoT preserves this central burden while reducing additional sources of noise. If code execution were allowed, a system could sometimes bypass this burden by externalizing the dependency structure into search, simulation, or optimization routines. LongCoT instead asks whether the model itself can carry that burden, testing long-horizon reasoning as a fundamental capability.

Each template is developed by a group of domain-experts, and we ensure that it reflects reasoning problems that are relevant and meaningful. We discuss how questions for each domain are constructed below.

### 3.2. Domains

We construct 10 templates of varying difficulty levels per domain and instantiate 50 questions per template. Final answers are directly verifiable [1] and inputs are short prompts with a median of 2K tokens. LongCoT consists of five domains: Mathematics, Chemistry, Chess, Computer Science,

---

[1] Several frontier models do not allow access to their reasoning traces, making outcome based verification important.

and Logic. Our templates are tuned to test domain-specific skills along with important properties (e.g. search/backtracking/state management/error correction) required to reason across long-horizons. We have three difficulty levels for questions: easy, medium, and hard, and we describe the process of subproblem selection, building meaningful long-horizon problems, realistic domain specific methods for composing tasks, and the different properties of long-horizon reasoning tested here. LongCoT tests a wide range of relevant reasoning problems (see Appendix A) to isolate and directly measure long-horizon reasoning capabilities.

**Mathematics.** We use compositional templates with atomic tasks drawn from Olympiad-level problems in Omni-Math (Gao et al., 2024) and HLE (Phan et al., 2025), covering algebra, number theory, combinatorics, probability, geometry, etc. Each node in the DAG is a competition problem with a short answer (integer, tuple, fraction, or finite set), and edges link problems so that inputs of later subproblems depend on solving earlier problems. We instantiate all four composition methods described in Subsection 3.1 (linear chains, multi-parent DAGs, conditionals, and forced backtracking) to test whether models can handle sequential dependencies, aggregate multiple inputs, and search over inputs to multi-step non-invertible functions it has to keep recomputing. Easy questions have 15 nodes, medium ones have 20–35, and hard problems have 40+.

**Chemistry.** We use compositional templates where atomic tasks involve structural and graph-based reasoning over molecules and proteins, e.g., identifying functional groups, substructure matching, stereochemistry analysis, counting ring types, and predicting reaction products or properties (Runcie et al., 2025). When composed into DAGs, these tasks form multi-step reasoning problems that mirror realistic chemistry workflows. Molecules are selected based on predicted properties, synthesized, and fed into subsequent reactions. Solving these problems requires tracking parallel synthesis paths until products combine, searching over a large space of molecules, and revising earlier predictions as new constraints emerge. Problem difficulty depends on DAG size and the structural properties/complexity of molecules being synthesized. Final answers are either SMILES strings or property prediction tuples. These are verified using cheminformatics tools (Landrum et al., 2006; Kim et al., 2025) and checked against USPTO forward synthesis data or the RCSB Protein Data Bank (Burley et al., 2024).

**Chess.** Our chess problems use procedural (implicit) templates, and involve problems such as best-move selection, mate-in-n, retrograde analysis (determining which moves could have led to a board state), knight pathfinding, and minimax games (opponent simulation). This provides a large, creative (Feng et al., 2025) set of hard but feasible long-

horizon problems where models must go beyond greedy approaches and evaluate multiple branches, backtrack from failed paths, and keep track of state such as visited squares, move legality, or positional constraints. We generate and verify problems using Stockfish, endgame tablebases, and exhaustive enumeration of board states. Difficulty scales with the required search depth or state-space size.

**Logic.** Our logic problems use procedural (implicit) templates that include constraint satisfaction puzzles (Sudoku), planning tasks (Sokoban, Blocks World), constrained pathfinding (Dungeon), and optimisation (Packaging Min Waste), all inducing large search spaces the model must navigate. For instance, CSPs test whether models can propagate constraints and predict downstream contradictions. Planning tasks test whether models can order sub-goals and recognize dead ends. Pathfinding tests whether models can track variables and plan possible routes over hundreds of steps. Across all problems, decisions made earlier heavily constrain later options, and errors can compound. Difficulty depends on grid size, the number of constraints, the optimal solution length, and the problem's algorithmic complexity.

**Computer Science.** Our computer science problems use both compositional and procedural templates to test whether models can precisely simulate deterministic processes over many steps. Problems include tracing program execution across loop iterations, scheduling instructions on parallel architectures (VLIW processor), executing graph algorithms (max-flow), finding efficient orderings for large matrix multiplication, simulating distributed memory systems, and stepping through type inference or compiler passes. Some templates require simulating a single complex algorithm across many iterations, and others compose multiple problems into DAGs. Each problem requires maintaining complex state across many steps, whether it's traversing syntax trees with variable substitutions in type inference, tracking partitioned data in distributed systems, or managing concurrent operations in scheduling. Difficulty scales with the number of iterations, the complexity of the simulated system, and the DAG depth, and problems are verified programmatically.

### 3.3. Benchmark composition

We present a distribution of topics for each domain along with some examples of the graph structures our problems explicitly or implicitly encode in Figure 3. Our benchmark contains 2,500 problems across five domains (500 per domain), generated through 10 realistic templates per domain with 50 parameter settings for each.[2] The benchmark con-

---

[2] Math templates use existing Olympiad-level questions that are tested for contamination before being composed. GPT 5.2 achieves 95.7% accuracy on isolated subproblems, yet accuracy on composed problems degrades well beyond what independent error compounding would predict (see Section 4.2).

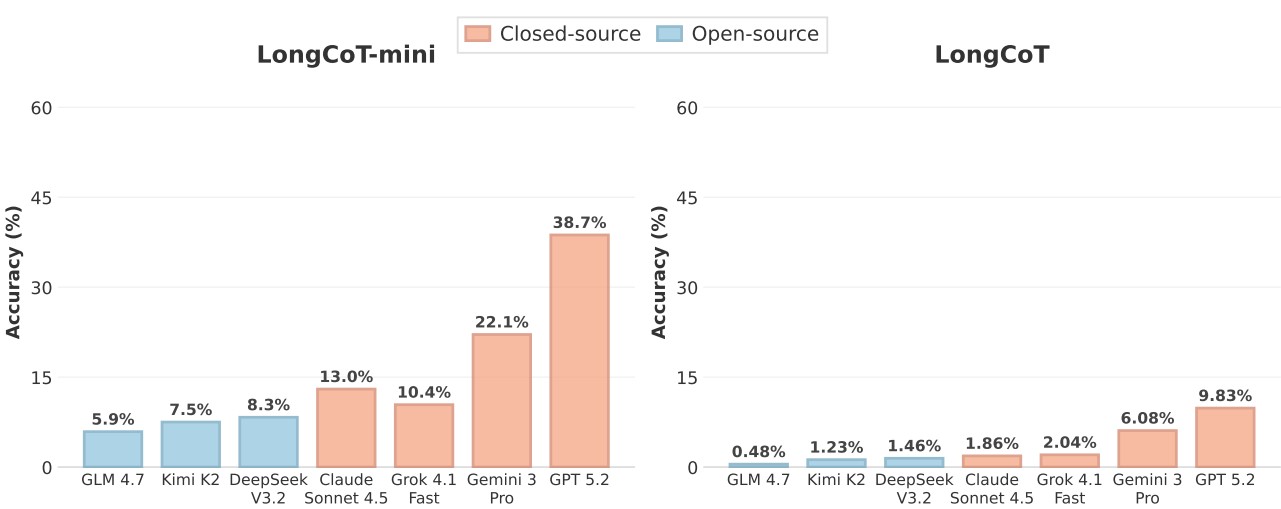

*Figure 4.* Main results on LongCoT-mini (left) and LongCoT (right). LongCoT is extremely challenging, with the best model (GPT 5.2) achieving only **9.83%** and open-source models near zero. LongCoT-mini differentiates performance across a wider range of models.

tains 500 easy problems (LongCoT-mini) that can be used to analyse the performance of non-frontier models. The remaining problems fall under our main benchmark LongCoT (750 medium, 1250 hard) and are designed to be challenging, but still fit within the output budget of frontier models.[3]

Final answers are drawn from large combinatorial spaces (e.g., integers, fractions, SMILES strings, move sequences), making random guessing ineffective. Incorrect intermediate values rarely yield correct final answers, enabling reliable outcome-only evaluation. Our templates are parameterized and composed by experts in ways distinct from general training data, minimizing the risk of pattern matching and requiring genuine long-horizon reasoning.

## 4. Evaluation

### 4.1. Frontier Models on LongCoT

We evaluate several frontier models on all 2,500 questions across our five domains. The frontier closed-source models we evaluate are GPT 5.2, Gemini 3 Pro, Claude 4.5 Sonnet[4], and Grok 4.1 Fast Reasoning. For open-source models, we evaluate DeepSeek V3.2 (DeepSeek-AI et al., 2025), Kimi K2 Thinking (Team et al., 2025b), and GLM 4.7 (Team et al., 2025a). These represent some of the most capable models as of mid-February 2026.

For all models, we enable CoT reasoning at the highest setting if available and allow reasoning up to the maximum

output limit permitted by providers (e.g., GPT 5.2 has a 128K limit). Models are evaluated in a single-shot setting with default temperature and top-p configurations. Most problems in our benchmark require reasoning over more than 50K tokens, which leads to high evaluation costs.[5] This prevents us from scaling up pass@k or self-consistency experiments. For API errors, we retry independently twice to ensure failures reflect model limitations rather than infrastructure issues. Closed-source models do not allow access to their CoT traces, restricting qualitative analysis to reasoning summaries or open-source model trajectories. Final answers are verified through sequential checks: RegEx on expected format, flexible RegEx on full responses if needed, and LLM-based extraction (GPT-5-mini) as a fallback. These answers are then manually verified for correctness.

LongCoT (2,000 medium and hard questions) is *extremely challenging for current frontier models* (Figure 4). We find that performance is uniformly low, with GPT 5.2 achieving the highest accuracy of **9.83%** followed by Gemini 3 Pro (6.08%) and Grok 4.1 Fast Reasoning (2.04%). Other models achieve near zero performance. Despite rapid progress on existing benchmarks, these results show that reliable long-horizon reasoning remains an open challenge for frontier models, with implications for tasks requiring sustained multi-step reasoning such as automatic science or enterprise work. It is also important to measure progress and differentiate smaller open-source models, for which we use LongCoT-mini (500 questions). These are easier long-horizon reasoning problems on which GPT 5.2 rises to 38.7%, and open-source models demonstrate competence, with Kimi K2 and

---

[3]For example, the average number of tokens to solve a single math subproblem on our benchmark is 1.5K, so solving 40 problems in a DAG should fit under the 128K limit of GPT 5.2.

[4]We are unable to evaluate Claude 4.5 Opus as its output token pricing is much higher than other closed-source frontier models.

[5]For example, the 3 best closed source frontier models cost an average of $13.66 USD per 1 million output tokens and $2.25 USD per 1 million input tokens (often around $1 USD per question).

DeepSeek V3.2 achieving 7.5% and 8.3% respectively. This confirms that our difficulty scaling method produces meaningful separation across a wide range of models.

## 4.2. Experimental Analysis

We analyze token usage on LongCoT (Figure 1). On average, GPT 5.2 uses 62,046 tokens per problem, which is higher than any other model. Figure 1 plots benchmark accuracy against the number of reasoning tokens used. Performance rises with token usage, and models that use *more of their available reasoning budget do markedly better*. Next, we analyze how long-horizon reasoning performance scales as the complexity of standard-DAG problems increases.

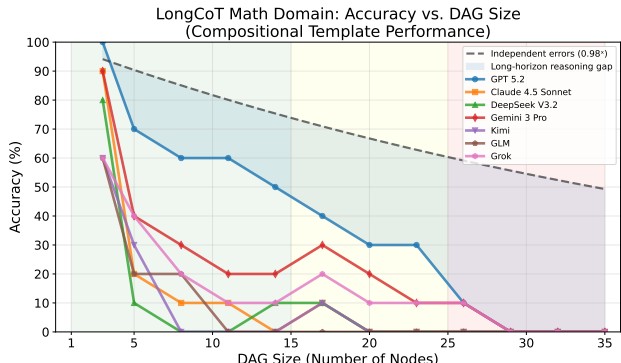

*Figure 6.* Accuracy falls as problem DAG sizes grow, inducing planning and execution difficulties before context windows saturate. Our problems increasingly differentiate model capabilities as horizon lengths grow. Under an independent error assumption (GPT 5.2 on Omni-Math), accuracy would be much higher than what we observe, highlighting issues with long-horizon reasoning.

Here, we use a *controlled setup* where we scale the number of nodes in composed Omni-Math questions from 1-35 and study how accuracy degrades for different models in Figure 6. All models show steep degradation as chain length increases, particularly beyond 15 nodes. Crucially, observed accuracy falls well below the independent error baseline (Motwani et al., 2025). This gap indicates that composed problems introduce additional failure modes that are absent when subproblems are solved in isolation. The gap between models widens at longer horizons, confirming that long-horizon problems *differentiate capabilities* more effectively than short ones. We further disentangle the effect of compositional reasoning from context length in App. C.

While LongCoT is designed specifically to measure CoT long-horizon reasoning abilities, we also evaluate it against the Recursive Language Models (RLM) framework (Zhang et al., 2025) with GPT 5.2, which recursively calls sub-agents to decompose problems and manage context. We evaluate RLM in two settings (Figure 7): a reasoning-only configuration where sub-agents solve problems without exe-

cuting code, aligned with LongCoT's goal of isolating reasoning capability, and the default RLM configuration where sub-agents may write and execute simulation code (with chess, rdkit, and other domain libraries available). In the reasoning-only setting, LongCoT's graph structured dependencies make problem decomposition harder, and context management between sub-agent calls sometimes loses track of important information, making planning and backtracking harder. Even with code execution enabled, performance improves primarily on implicit domains where substantial parts of the dependency structure can be externalized to code (Logic: 68.3%, Chess: 30.6%), while explicit compositional domains remain more challenging. Zhang et al. (2025) show how specific prompts for decomposition can improve RLM performance further on LongCoT-mini.

## 4.3. Qualitative Analysis

We analyze the reasoning traces of open source models to show how much reasoning is spent on solving the problem, planning steps, setting up the problem, verifying steps, dead-ends, and backtracking. In Figure 8, we present two examples, one on LongCoT-Logic and one on LongCoT-Math to show how different reasoning traces look for each model and domain. This emphasizes the importance of having five domains with distinct structures as part of LongCoT.

In general, we observe the following fundamental issues in long-horizon reasoning capabilities. Poor early planning commits models to inefficient strategies, and errors compound across time as incorrect values propagate through dependent steps. Models exhaust their effective context and resort to guessing, or give up prematurely on problems they can sometimes solve with a fresh attempt. Finally, models often fail to backtrack and explore alternative paths. We provide detailed trajectory analysis in Appendix C.

## 5. Discussion

LongCoT is designed to measure the capability of models to sustain a correct, coherent chain of dependent reasoning steps over tens to hundreds of thousands of tokens. This addresses a gap in existing evaluations, which tend to interleave long-horizon reasoning ability with separate agentic skills and scaffolding properties such as tool use, retrieval, external memory, and task-specific orchestration. The difficulty of LongCoT comes not from requiring esoteric knowledge or advanced reasoning abilities, but because it forces models to plan, maintain state, and control errors while the reasoning trace becomes long enough for drift and compounding mistakes to accumulate. Our templates explicitly target these abilities by ensuring that local steps are tractable for current models, so frontier accuracy being $< 10\%$ indicates that the limiting factors are long-horizon

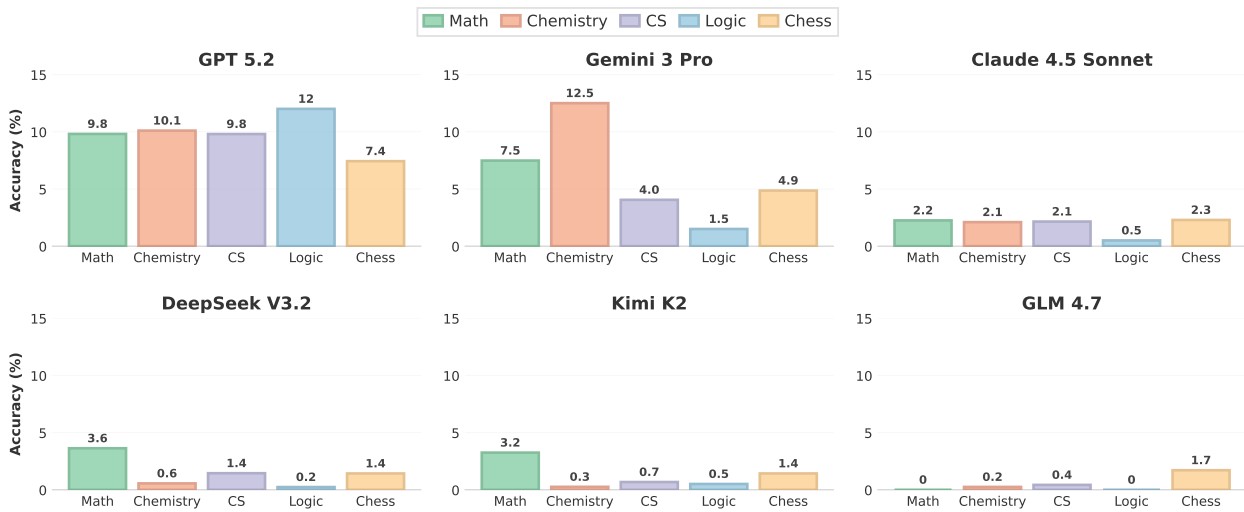

*Figure 5.* LongCoT domain-specific results are mostly stable across all five domains for a given model. These findings comport with the design goals of LongCoT: rather than deep domain knowledge, LongCoT success demands the ability to reason over long horizons.

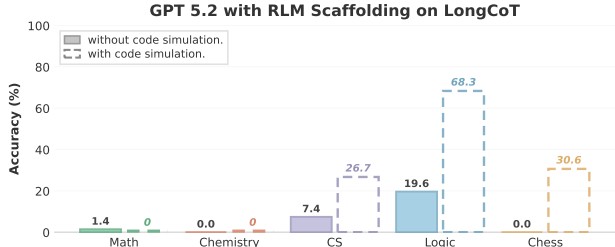

*Figure 7.* RLM evals. GPT-5.2 RLM tested in reasoning-only (solid) and code simulation (dashed) settings. Without sub-agents calling tools, RLM does not outperform GPT 5.2 alone. With code simulation, RLM delivers gains on implicit domains, where substantial parts of the dependency structure can be offloaded to programmatic search routines. See Appendix C for analysis.

specific skills. This is crucial for autonomous deployment, as agents may be able to offload certain computations, but still depend on the model to maintain an internal state of goals, constraints, partial results, etc., over long trajectories. When this core reasoning fails and the dependency structure cannot be externalized, scaffolding and tool use alone are insufficient to repair it. Many of the most ambitious applications of AI, from scientific discovery to drug design to complex engineering, will require reliable reasoning over extended horizons as one of several necessary capabilities. LongCoT provides an important benchmark for progress on this axis, as improvements that raise performance across our diverse domains would provide evidence of more reliable long-horizon reasoning, and would likely contribute to more effective agents. This suggests a partial path forward (see Section D) via developing training and inference methods that explicitly target long-horizon stability and using LongCoT as a direct, verifiable measure of whether they genuinely improve long-horizon reasoning.

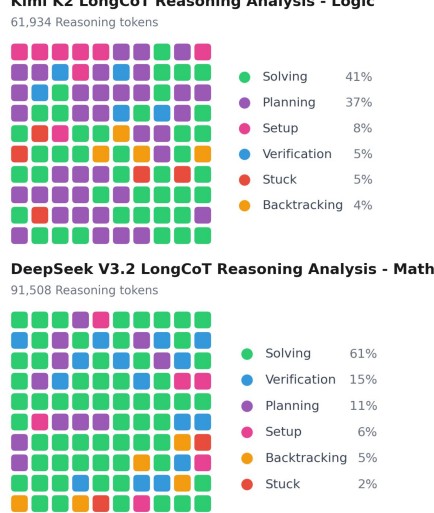

*Figure 8.* Reasoning trace analysis. The distribution of reasoning spent across behaviors varies substantially by domain and model. Each cell represents 1% of the reasoning trace, read left-to-right, top-to-bottom (see Appendix C for more analysis).

## Impact Statement

This paper presents work whose goal is to advance the field of Machine Learning. It aims to serve as a benchmark for an important direction related to improving language models and autonomous agents. There are potential societal consequences of our work, as improvements in language models also lead to increased potential for misuse. Overall, being able to measure the long-horizon reasoning skills of a model provides an additional measure of their capabilities at deployment time.

## Acknowledgments

Prepared by LLNL under Contract DE-AC52-07NA27344 and supported by the LLNL-LDRD Program under Projects No. 24-ERD-058, 24-SI-008, and 26-ERD-019 (LLNL-CONF-2017880). This manuscript has been authored by Lawrence Livermore National Security, LLC under Contract No. DE-AC52-07NA27344 with the U.S. Department of Energy. The United States Government retains, and the publisher, by accepting the article for publication, acknowledges that the United States Government retains a non-exclusive, paid-up, irrevocable, world-wide license to publish or reproduce the published form of this manuscript, or allow others to do so, for United States Government purposes.

We thank the Royal Academy of Engineering, Cooperative AI Foundation, and Schmidt Sciences for their support.

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

# Appendix

# A. Per-domain Results

This section provides, for each domain, descriptions of problem templates and detailed LongCoT/LongCoT-mini benchmark performances. The performance results are reported for each problem template and model. Overall, LongCoT-mini is substantially cheaper to run (approximately $150 for GPT 5.2, compared to approximately $2,500 on the full LongCoT benchmark) while still providing high signal across model classes.

## A.1. Logic

**BlocksWorld.** In the BlocksWorld problem, the model is presented with three stacks of numbered blocks in an initial configuration and must determine a sequence of moves to rearrange them into a specified goal configuration. Each move consists of taking the topmost block from one stack and placing it either on an empty position or on top of another stack. Only the topmost block of any stack can be moved at any time. The model must output a valid sequence of moves, where each move is specified as a triple containing the block identifier, the source stack, and the destination stack. Success requires finding a feasible plan that transforms the initial state into the goal state while respecting the constraint that only top blocks can be manipulated. This problem tests *planning* capabilities, as the model must organise a sequence of moves to efficiently reach the goal. It requires *exploration* to evaluate different move sequences, and *self-evaluation and backtracking* when attempted moves lead to configurations from which the goal becomes unreachable.

**Dungeon.** The Dungeon problem is a grid-based pathfinding challenge with health management mechanics. The model receives a two-dimensional grid where each cell contains a health delta value that can be positive or negative. Starting from the top-left corner, the model must determine the minimum initial health required to reach the bottom-right corner while keeping health strictly positive throughout the journey. Movement is restricted to only rightward or downward directions, one cell at a time. Upon entering each cell, the player's health changes by that cell's delta value. The model must compute and output a single integer representing the minimum starting health that guarantees survival along some valid path to the destination. This problem tests *context management and recall*, as the model must track cumulative health values and optimal subproblem solutions across many cells throughout the computation.

**PackagingMinWaste.** In the PackagingMinWaste problem, the model must optimise a supplier selection and bin-packing scenario. Given a set of packages with various sizes and multiple suppliers each offering different box sizes, the model must choose exactly one supplier and pack all packages into boxes from that supplier while minimising total wasted space. Waste is defined as the sum of unused space in each box, calculated as the difference between box size and package size. Each package must fit in exactly one box, and a package of a given size can only be placed in a box of equal or larger size. The model outputs the minimum total waste modulo $10^9 + 7$, or $-1$ if no valid supplier exists. This problem tests *context management and recall* to track package assignments and cumulative waste across many items and suppliers.

**RandomHanoi.** The RandomHanoi problem extends the Tower of Hanoi puzzle to arbitrary initial and goal configurations. The model is given three stacks containing discs of different sizes, where larger size indices correspond to larger discs. The task is to find a sequence of moves that transforms the initial disc arrangement into the goal arrangement. Unlike BlocksWorld, this problem enforces the Tower of Hanoi constraint: a disc can only be placed on an empty stack or on top of a larger disc. The model must output a valid move sequence where each move specifies which disc to move and between which stacks, ensuring that the size ordering constraint is maintained after every move. This problem tests *planning* capabilities due to the interdependencies between disc positions. It requires *exploration* to search through possible move sequences, and *self-evaluation and backtracking* since invalid moves violate size constraints and may create states from which the goal is difficult to reach.

**Sokoban.** Sokoban is a grid-based puzzle where the model controls a player character who must push boxes onto designated goal positions. The grid contains walls that block movement, boxes that can be pushed but not pulled, and goal locations where boxes must be placed. The player can move in four cardinal directions (up, down, left, right), and pushing a box requires moving into it when there is empty space on the opposite side. Boxes cannot be pushed into walls or other boxes. The model must output a string of movement commands that successfully places all boxes on goal positions, demonstrating spatial reasoning and planning capabilities. This problem tests *planning*, as pushing boxes into corners or against walls creates irreversible dead ends. It requires *exploration* to search for valid solution paths, and *self-evaluation and backtracking* to recognise and recover from mistakes.

**Sudoku.** The Sudoku problem presents the model with a partially filled grid that must be completed according to standard Sudoku rules. The output is the complete solved grid represented as a list of rows. Difficulty is controlled by the density of pre-filled cells, with harder puzzles providing fewer initial clues. This problem tests *context management and recall* to track constraints across rows, columns, and subgrids simultaneously. It requires *exploration* to try candidate values when multiple options exist, and *self-evaluation and backtracking* when digit placements lead to constraint violations discovered only after subsequent fills.

**TrapezoidCounting.** In the TrapezoidCounting problem, the model receives a set of points in two-dimensional Cartesian coordinates and must count the number of distinct trapezoids that can be formed by selecting any four points. A valid trapezoid is defined as a convex quadrilateral with at least one pair of parallel sides, where parallelism is determined by equal slopes. The four chosen points must form a proper convex quadrilateral without any three points being collinear and without any concave angles. The model must output a single integer representing the total count of distinct trapezoid configurations, requiring geometric reasoning and systematic enumeration. This problem primarily tests *context management and recall* to track geometric relationships and running counts across many point combinations.

**WizardsTotalStrength.** The WizardsTotalStrength problem is an algorithmic challenge involving array computations with modular arithmetic. The model receives an array of integer strength values and must compute a sum over all contiguous subarrays. For each subarray, the contribution to the total is defined as the product of the minimum element and the sum of all elements in that subarray. The model must sum these contributions across every possible contiguous subarray and return the result modulo $10^9 + 7$. This problem tests the model's ability to recognise efficient algorithmic patterns, as naive enumeration of all subarrays would be computationally prohibitive for large inputs. This problem tests *context management and recall* to track intermediate sums, minimums, and running totals across numerous subarrays.

*Table 2.* Model accuracy (%) on Logic tasks by template and difficulty

| Template | Difficulty | GPT5.2 | Gemini 3 Pro | Claude 4.5 Sonnet | Deepseek V3.2 | Grok 4.1 Fast | GLM | Kimi |
|---|---|---|---|---|---|---|---|---|
| BlocksWorld | Easy | 60.0 | 0.0 | 0.0 | 0.0 | 0.0 | 0.0 | 0.0 |
| | Medium | 16.0 | 0.0 | 0.0 | 0.0 | 0.0 | 0.0 | 0.0 |
| | Hard | 4.0 | 0.0 | 0.0 | 0.0 | 0.0 | 0.0 | 0.0 |
| Dungeon | Easy | 53.3 | 0.0 | 0.0 | 0.0 | 0.0 | 0.0 | 0.0 |
| | Medium | 12.0 | 4.0 | 4.0 | 0.0 | 0.0 | 0.0 | 0.0 |
| | Hard | 0.0 | 8.0 | 0.0 | 0.0 | 0.0 | 0.0 | 8.0 |
| PackagingMinWaste | Easy | 66.7 | 6.7 | 0.0 | 6.7 | 0.0 | 0.0 | 0.0 |
| | Medium | 24.0 | 4.0 | 0.0 | 0.0 | 0.0 | 0.0 | 0.0 |
| | Hard | 4.0 | 0.0 | 0.0 | 0.0 | 0.0 | 0.0 | 0.0 |
| RandomHanoi | Easy | 40.0 | 13.3 | 0.0 | 0.0 | 0.0 | 0.0 | 0.0 |
| | Medium | 20.0 | 4.0 | 0.0 | 0.0 | 0.0 | 0.0 | 0.0 |
| | Hard | 0.0 | 4.0 | 0.0 | 0.0 | 0.0 | 0.0 | 0.0 |
| Sokoban | Easy | 40.0 | 10.0 | 0.0 | 0.0 | 10.0 | 0.0 | 10.0 |
| | Medium | 5.0 | 0.0 | 0.0 | 0.0 | 0.0 | 0.0 | 0.0 |
| | Hard | 4.0 | 0.0 | 0.0 | 0.0 | 0.0 | 0.0 | 0.0 |
| Sudoku | Easy | 50.0 | 0.0 | 30.0 | 0.0 | 0.0 | 0.0 | 0.0 |
| | Medium | 48.0 | 0.0 | 4.0 | 0.0 | 0.0 | 0.0 | 0.0 |
| | Hard | 0.0 | 0.0 | 0.0 | 0.0 | 0.0 | 0.0 | 0.0 |
| TrapezoidCounting | Easy | 53.3 | 26.7 | 0.0 | 13.3 | 0.0 | 6.7 | 13.3 |
| | Medium | 16.0 | 0.0 | 0.0 | 4.0 | 0.0 | 0.0 | 0.0 |
| | Hard | 0.0 | 0.0 | 0.0 | 0.0 | 0.0 | 0.0 | 0.0 |
| WizardsTotalStrength | Easy | 60.0 | 0.0 | 0.0 | 0.0 | 0.0 | 0.0 | 0.0 |
| | Medium | 40.0 | 0.0 | 0.0 | 0.0 | 0.0 | 0.0 | 0.0 |
| | Hard | 0.0 | 0.0 | 0.0 | 0.0 | 0.0 | 0.0 | 0.0 |
| **LongCoT-mini** | | **53.6** | **7.3** | **2.7** | **2.7** | **0.9** | **0.9** | **2.7** |
| **LongCoT** | | **12.0** | **1.5** | **0.5** | **0.3** | **0.0** | **0.0** | **0.5** |

## A.2. Computer Science

**Hindley-Milner Type Inference.** In the Hindley-Milner type inference problem, the model must execute Algorithm W on a functional program containing lambda expressions, let-bindings, pairs, and primitive operations. The model traces through the deterministic unification process, recording each type variable binding as it occurs. The model must report the principal type schemes for specific let-bindings, the final result type, the total number of bindings in the trace, and specific bindings at designated checkpoint positions in a precise prefix format. All type schemes must be normalised with proper quantifier ordering and human-readable variable names. This problem tests *context management and recall*, as the model must track type variable bindings, substitution contexts, and unification state across hundreds of inference steps.

**Max-Flow Min-Cut Gauntlet.** The Max-Flow Min-Cut Gauntlet presents a multi-round challenge on a directed graph with edge capacities. In each round, the model must compute the maximum flow using the Edmonds-Karp algorithm with precise BFS neighbour ordering, identify the minimum cut set of vertices reachable from the source, extract a dominant route from source to sink using DFS with specific backtracking rules, and identify damage and repair edges according to tie-breaking criteria. The state after each round determines parameters for the next round through modular arithmetic updates to capacities and terminal vertices. The model must output the final edge capacities after all rounds are complete. This problem tests *context management and recall* to track flow values, residual capacities, and graph state across multiple rounds where each round's output influences the next.

**Chained Scheduling Simulation.** The Chained Scheduling problem presents eight interconnected scheduling subproblems with data dependencies between them. Each subproblem uses a different scheduling algorithm, including EDF (Earliest Deadline First), list scheduling on DAGs with critical-path priorities, round-robin with time quanta, and SJF (Shortest Job First) with release times. Later problems use outputs from earlier problems to parameterise their inputs, creating cascading dependencies where errors propagate. The model must correctly implement each scheduling algorithm with precise tie-breaking rules, track job states and completion times, and compute the answer for each subproblem that feeds into subsequent computations. This problem tests *context management and recall* to track job execution states, algorithm-specific data structures, and inter-problem dependencies across the entire chain of eight subproblems.

**Turing Machine Simulation.** The Turing Machine Simulation requires the model to simulate a 3-tape deterministic Turing machine across eight dependent problems. The machine operates on windows of a base bitstring, with later simulations using outputs from earlier ones to derive window offset and size parameters. For each problem, the model simulates the specified number of steps using first-match rule semantics, managing head positions on three bi-infinite tapes. The final state is encoded as a scalar combining the state identifier, head positions modulo a prime, and symbols under each head. Each answer feeds into the next problem's parameters, requiring precise execution throughout. This problem tests *context management and recall* to track tape contents, head positions, machine state, and the chained parameter dependencies across all eight simulation problems.

**Program Simulation** In the Program Simulation problem, the model must exactly execute a short Python-like imperative loop program over scalar integers. After completing the simulation, the model answers questions drawn from a pool of trace-dependent questions, such as finding the first/last iteration where a predicate holds (e.g., `x>y`), counting how many iterations exceed a substituted threshold, locating the start of a consecutive span satisfying a condition, reporting extrema over the run, or outputting a state-derived value (e.g., `x+y`) at a specified iteration.

**VLIW Scheduling** This template describes a very long instruction word (VLIW) computer architecture with many ALU slots. A simple assembly program is provided (in a very simple, made-up assembly language) and the language model is tasked with finding ideal execution schedules for the program on the VLIW architecture. Questions about the scheduling are parameterized by ALU slot counts and cycle counts for each instruction type.

**Matmul** In the Matmul problem, the model is given a sequence of matrices with explicit dimensions and must solve the matrix-chain multiplication task by selecting the parenthesization that minimises total arithmetic cost, assuming standard dense multiplication with cost proportional to $a \cdot b \cdot c$ for multiplying an $a{\times}b$ matrix by a $b{\times}c$ matrix. The model must output the optimal fully-parenthesized expression in a strict format, then compute several traceable properties of that decomposition: the total floating-point operation count for the chosen order, the maximum parenthesis nesting depth, a span-based parenthesis distance metric after simplifying trivial left-to-right chains, and a final scalar that combines these quantities.

**IR Optimization** In the IR Optimization problem, the model is given an LLVM IR function (typically `@main`) and must answer a chained set of questions. Queries mix static IR analysis (counts, locating the $k$-th instruction's block, opcode string

properties, and line numbers) with single-pass transformation reasoning, where a pass chosen from a fixed list (e.g., `dce`, `adce`, `simplifycfg`, `mem2reg`) is applied and the model re-counts post-pass opcodes and instruction totals.

**Distributed Memory Tracing** In this template the model is tasked with tracing data values in a distributed memory parallel program. More specifically, pseudocode for a message passing interface (MPI) program is given and questions are asked about which value is on which process. The questions sometimes shift or modify the values and inquire about the impact this has on the algorithm. This question template tests a model's ability to track values in distributed memory programs.

**Backpropagation** These questions, similar to distributed memory tracing, involve tracing memory values during program execution, except that it involves tracing float values during the backpropagation algorithm. Initial conditions and network layers are provided to the model and questions are asked about data values and how they would change if certain interventions were made.

*Table 3.* Model accuracy (%) on CS tasks by template and difficulty

| Template | Difficulty | GPT5.2 | Gemini 3 Pro | Claude 4.5 Sonnet | Deepseek V3.2 | Grok 4.1 Fast | GLM | Kimi |
|---|---|---|---|---|---|---|---|---|
| HindleyMilner | Easy | 38.0 | 0.0 | 4.0 | 0.0 | 2.0 | 0.0 | 0.0 |
| Scheduling | Medium | 22.0 | 0.0 | 0.0 | 0.0 | 0.0 | 0.0 | 2.0 |
| TuringMachine | Medium | 10.0 | 0.0 | 0.0 | 0.0 | 0.0 | 0.0 | 0.0 |
| MaxFlowMinCut | Hard | 0.0 | 0.0 | 0.0 | 0.0 | 0.0 | 0.0 | 0.0 |
| Program Sim. | Easy | 45.0 | 45.0 | 20.0 | 30.0 | 20.0 | 25.0 | 35.0 |
| | Medium | 16.7 | 13.3 | 10.0 | 0.0 | 10.0 | 0.0 | 3.3 |
| | Hard | 10.0 | 6.7 | 3.3 | 0.0 | 0.0 | 0.0 | 0.0 |
| VLIW Sched. | Easy | 25.0 | 30.0 | 10.0 | 40.0 | 25.0 | 20.0 | 25.0 |
| | Medium | 10.0 | 16.7 | 3.3 | 10.0 | 13.3 | 0.0 | 0.0 |
| | Hard | 6.7 | 3.3 | 0.0 | 0.0 | 0.0 | 0.0 | 0.0 |
| Matmul | Easy | 80.0 | 40.0 | 20.0 | 15.0 | 40.0 | 10.0 | 35.0 |
| | Medium | 26.7 | 13.3 | 13.3 | 6.7 | 20.0 | 6.7 | 3.3 |
| | Hard | 6.7 | 10.0 | 3.3 | 6.7 | 6.7 | 0.0 | 0.0 |
| IR Opt | Easy | 55.0 | 25.0 | 30.0 | 10.0 | 20.0 | 30.0 | 20.0 |
| | Medium | 10.0 | 16.7 | 6.7 | 6.7 | 6.7 | 0.0 | 3.3 |
| | Hard | 6.7 | 0.0 | 0.0 | 0.0 | 0.0 | 0.0 | 0.0 |
| Dist. Memory Trace | Easy | 45.0 | 40.0 | 20.0 | 40.0 | 10.0 | 10.0 | 20.0 |
| | Medium | 6.7 | 3.3 | 6.7 | 3.3 | 3.3 | 3.3 | 0.0 |
| | Hard | 3.3 | 0.0 | 0.0 | 0.0 | 6.7 | 0.0 | 0.0 |
| Backprop. | Easy | 55.0 | 25.0 | 25.0 | 30.0 | 20.0 | 10.0 | 5.0 |
| | Medium | 20.0 | 10.0 | 3.3 | 0.0 | 0.0 | 0.0 | 0.0 |
| | Hard | 13.3 | 0.0 | 0.0 | 0.0 | 3.3 | 0.0 | 0.0 |
| **LongCoT-mini** | | **44.4** | **17.1** | **12.4** | **13.9** | **12.1** | **8.6** | **11.4** |
| **LongCoT** | | **9.7** | **3.0** | **1.4** | **1.1** | **2.6** | **0.2** | **0.4** |

## A.3. Chemistry

A prerequisite to almost all reaction sub-questions is understanding SMILES strings, as molecules are provided as SMILES strings in almost all cases.

In the chemistry questions, most subquestions will select a molecule that will be used in future subquestions. For all but Reaction Prediction sub-questions, they do this by either ranking and picking molecule by a certain property of value, or using such a property or value as a threshold to pick one molecule or another.

**Molecule Properties.** Some sub-questions relate to general molecular properties, such as molecular weight, hydrogen counts and heavy atom counts, Lipsinki's rule of 5, aromaticity, chirality, and so on. This tests a models knowledge of these basic properties of a molecule. The sub-questions typically pick a certain molecule from a list based on which one has the highest or n-th highest of a certain property, or picks from two molecules if a property of a molecule is more than some

threshold. When we compare quantities for several molecules at once, the model must reliably derive and compare the properties rather than estimating them with surface level heuristics. Some sub-questions will also ask for multiples of these properties at once, sub selecting from a list based on how their properties compare to the reference properties. This opens the model to more chances to backtrack and rethink, if they end up selecting no molecules.

**Reaction Prediction.** Reactions between molecules are a fundamental part of chemistry. In these sub-questions, the model is asked to predict the product of a reaction with at least two reactants that has a certain condition (temperature and solvent), and certain reactants that may be specified by previous sub-questions. Here we allow the model to use those conditions, or whatever conditions it thinks is more appropriate. To do reaction prediction well, a model must understand topics like functional group compatibility, reaction mechanisms, and competing pathways. And when multiple reactants are chosen by previous sub questions, this opens the model up to allow backtracking: Two incompatible reactants can imply that a model made an incorrect choice in a previous sub-question, spurring them to revisit the previous sub-question.

**Molecular Topology.** These sub-questions evaluates a model's understanding of molecular structure by asking about topological distances between atoms. Examples include tological diameter (the maximal shortest path between any two atoms), Wiener index (sum of all pairwise distances), atom eccentricity (max distance between a certain atom and all others), and so on. Answering these problems correctly requires the model to reason over the full molecular graph, enumerating over the necessary pairs of atoms, and count the length of the path between them, instead of using heuristics. Understanding molecular topology can strongly influence physical properties and reactivity.

**Substructures.** These subquestions ask if a certain functional group or substructure is present or not in a molecule. Functional groups and substructures are drawn randomly from a list of common substructures and RDKit fragment descriptors (like amides, carboxylic acid, furan ring). This requires the model to recognize and understand common motif names. Knowing these common motifs and functional groups is a major component to reaction chemistry, and many other chemistry domains in general, functional groups determine a lot of chemical behavior.

**Molecule Representations.** A few sub-questions ask to identify a molecule that has a different SMILES string but represents the same underlying molecule. This tests basic understanding of SMILES. Some questions use an adjacency matrix plus a list of atom types to identify molecules, instead of a SMILES string. Solving these problems requires invariance to representational choices and an understanding of molecular equivalence, rather than memorization of a single representation. This capability is essential for reasoning across heterogeneous chemical data sources.

**Ring Analysis.** These sub-questions ask about rings properties, such as counting the number of a certain type of ring, getting the largest ring, and getting the smallest set of smallest rings. Cyclic structures in molecules are central to many areas of chemical reasoning.

**Proteins Substructures.** These problems test a model's ability to perform multi-step reasoning over protein sequence, geometry, and structure. The model must analyze protein backbone geometry, predict secondary structure from geometric features, and infer the identities of missing residues before performing a structure-aware ranking of selected residues. Reasoning over geometric and structural features mirrors an inverse folding task where the objective is to derive an amino acid sequence from structure. The model predicts missing residues based on local structural context (compactness and backbone angles) and predicted secondary structure, demonstrating whether the model can reason backward from 3D geometry to sequence identity.

## A.4. Chess

**Templates 1&6: Next Best Moves.** These templates test classical chess positional and tactical evaluation. Given a board position in FEN notation, the model must identify the strongest continuation(s). Template 1 requires ranking the top three moves, while Template 6 focuses on finding the single optimal move in a puzzle context. These problems assess the ability to evaluate piece activity, king safety, material balance, and tactical threats; core skills that require integrating pattern recognition with the forward calculation of variations.

**Template 2: Reverse Game Reconstruction.** This is a constraint-satisfaction + causality task: the model must reason backward and forward to build a legal, alternating SAN sequence consistent with the final board, the side to move, and the embedded "must-include" move fragments. It is uniquely long-horizon because early choices (piece routes, captures, checks, interpositions) constrain later legality, and SAN disambiguation/check markers force the model to maintain an exact board state throughout—one mistake anywhere invalidates the entire reconstruction.

*Table 4.* Model accuracy (%) on Chemistry tasks by template and difficulty

| Template | Difficulty | GPT5.2 | Gemini 3 Pro | Claude 4.5 Sonnet | Deepseek V3.2 | Grok 4.1 Fast | GLM | Kimi |
|---|---|---|---|---|---|---|---|---|
| | Easy | 46.0 | 54.0 | 38.0 | 30.0 | 48.0 | 20.0 | 22.0 |
| | Easy | 34.0 | 42.0 | 28.0 | 10.0 | 32.0 | 6.0 | 10.0 |
| | Medium | 14.0 | 16.0 | 4.0 | 0.0 | 12.0 | 0.0 | 2.0 |
| | Medium | 10.0 | 28.0 | 12.0 | 2.0 | 8.0 | 0.0 | 2.0 |
| Molecules | Medium | 8.0 | 16.0 | 6.0 | 2.0 | 0.0 | 0.0 | 2.0 |
| | Hard | 2.0 | 0.0 | 0.0 | 0.0 | 0.0 | 0.0 | 0.0 |
| | Hard | 0.0 | 0.0 | 0.0 | 0.0 | 0.0 | 0.0 | 0.0 |
| | Hard | 6.0 | 6.0 | 2.0 | 0.0 | 0.0 | 0.0 | 0.0 |
| Protein | Medium | 28.0 | 30.0 | 2.0 | 0.0 | 4.0 | 2.0 | 0.0 |
| | Hard | 14.0 | 16.0 | 0.0 | 0.0 | 0.0 | 0.0 | 0.0 |
| **LongCoT-mini** | | **40.0** | **48.0** | **33.0** | **20.0** | **40.0** | **13.0** | **16.0** |
| **LongCoT** | | **10.2** | **14.0** | **3.3** | **0.5** | **3.0** | **0.3** | **0.8** |

**Template 3&8: Move Sequence Simulation.** These templates test accurate state tracking over extremely long move sequences (hundreds of UCI moves). Template 3 requires outputting the final FEN after executing all moves, while Template 8 adds a tactical evaluation component by then asking for the best next move. These problems stress-test the model's ability to maintain perfect accuracy through extended sequential operations—a single misapplied move propagates errors throughout the remainder.

**Template 4: Knight Traveling Salesman.** This template poses a combinatorial optimization challenge on an oversized 100×100 board. The model must find the minimum-move path for a knight to visit all specified target squares. This adapts the classic traveling salesman problem to knight-move geometry, requiring both pathfinding between points and global route optimization. The large board size and numerous targets make exhaustive search infeasible, demanding heuristic or algorithmic approaches.

**Template 5: Minimax Knight Capture Game** This template combines game theory with chess mechanics in a two-player optimization scenario. Alice and Bob alternately select pawns to capture with a knight, with Alice maximizing and Bob minimizing the total moves. This requires computing knight distances to all pawns and reasoning through a minimax game tree. The problem tests multi-step adversarial reasoning, where optimal play depends on anticipating the opponent's counter-strategy.

**Template 7: Simultaneous Movement Combinatorics** This template explores a non-standard chess variant where multiple pieces move simultaneously toward target cells provided as inputs. The task is to count valid move combinations where no two pieces ever occupy the same square mid-transit. This requires modeling piece trajectories as paths through time-space and checking all pairwise intersection possibilities, a combinatorial explosion that tests systematic enumeration and collision detection logic.

**Template 9: Maximum Rook Placement with Obstacles** This is large-scale optimization under structured constraints, essentially a maximum bipartite matching / maximum independent set in a rook-attack grid after deletions. It is interesting for long-horizon reasoning since the deletions create irregular feasible regions where local placement intuition breaks; the correct maximum comes from global structure (rows/columns connectivity after blocked cells), pushing the model toward systematic decomposition and algorithmic reasoning rather than tactical chess knowledge.

**Template 10: Constrained Knight Pathfinding** This template requires navigating a knight from a starting position to the goal while avoiding squares attacked by enemy pieces. Unlike simple BFS pathfinding, the model must first compute the full attack coverage of rooks, bishops, and knights, creating a complex obstacle map. This tests both chess rule knowledge (how each piece attacks) and pathfinding under constraints, with the possibility that no valid path exists depending on enemy placement.

*Table 5.* Model accuracy (%) on Chess tasks by template and difficulty

| Template | Difficulty | GPT 5.2 | Gemini 3 Pro | Claude 4.5 Sonnet | Deepseek V3.2 | Grok 4.1 Fast | GLM | Kimi |
|---|---|---|---|---|---|---|---|---|
| Template 1 | Hard | 0.0 | 0.0 | 0.0 | 0.0 | 0.0 | 0.0 | 0.0 |
| Template 2 | Medium | 16.0 | 2.0 | 0.0 | 0.0 | 0.0 | 0.0 | 0.0 |
| Template 3 | Easy | 54.0 | 8.0 | 2.0 | 0.0 | 0.0 | 0.0 | 0.0 |
| Template 4 | Hard | 0.0 | 8.0 | 4.0 | 0.0 | 0.0 | 0.0 | 0.0 |
| Template 5 | Hard | 4.0 | 4.0 | 0.0 | 2.0 | 0.0 | 2.0 | 6.0 |
| Template 6 | Easy | 18.0 | 40.0 | 18.0 | 4.0 | 6.0 | 8.0 | 12.0 |
| Template 7 | Easy | 38.0 | 0.0 | 0.0 | 0.0 | 0.0 | 0.0 | 0.0 |
| Template 8 | Medium | 32.0 | 14.0 | 2.0 | 6.0 | 0.0 | 4.0 | 0.0 |
| Template 9 | Hard | 0.0 | 0.0 | 0.0 | 0.0 | 0.0 | 0.0 | 0.0 |
| Template 10 | Hard | 0.0 | 6.0 | 10.0 | 2.0 | 0.0 | 6.0 | 4.0 |
| **LongCoT-mini** | | **36.7** | **16.0** | **6.7** | **1.3** | **2.0** | **2.7** | **4.0** |
| **LongCoT** | | **7.4** | **4.9** | **2.3** | **1.4** | **0.0** | **1.7** | **1.4** |

## A.5. Math

We construct math questions based on five templates: Linear, DAG, Conditional, Backtrack, and DAG-First. The linear templates simply place math problems in order with a linear dependence from one question to the next. Similarly, the DAG template creates dependencies between math problems, but in the form of a DAG structure. The conditional and backtrack templates introduce two separate types of DAG structures; conditionals create nodes in the DAG that branch conditionally based on a previous nodes value and backtracking nodes rely on solving inverse functions of sub-dags. Finally, the DAG-First template is the same as DAG, but present the problem structure (as a DAG) first before listing all the math problems that make up the nodes.

*Table 6.* Model accuracy (%) on Math tasks by DAG template and difficulty

| Template | Difficulty | GPT5.2 | Gemini 3 Pro | Claude 4.5 Sonnet | Deepseek V3.2 | Grok 4.1 Fast | GLM | Kimi |
|---|---|---|---|---|---|---|---|---|
| Linear | Easy | 65.0 | 40.0 | 20.0 | 20.0 | 25.0 | 20.0 | 15.0 |
| | Medium | 36.7 | 10.0 | 13.3 | 16.7 | 16.7 | 0.0 | 13.3 |
| | Hard | 6.0 | 4.0 | 4.0 | 4.0 | 4.0 | 0.0 | 2.0 |
| DAG | Easy | 55.0 | 25.0 | 5.0 | 25.0 | 10.0 | 5.0 | 0.0 |
| | Medium | 20.0 | 20.0 | 3.3 | 10.0 | 16.7 | 0.0 | 13.3 |
| | Hard | 4.0 | 4.0 | 2.0 | 0.0 | 6.0 | 0.0 | 0.0 |
| Conditional | Easy | 35.0 | 5.0 | 5.0 | 10.0 | 0.0 | 5.0 | 5.0 |
| | Medium | 26.7 | 10.0 | 0.0 | 3.3 | 6.7 | 0.0 | 6.7 |
| | Hard | 2.0 | 2.0 | 0.0 | 0.0 | 0.0 | 0.0 | 0.0 |
| Backtrack | Easy | 30.0 | 20.0 | 0.0 | 0.0 | 15.0 | 0.0 | 0.0 |
| | Medium | 6.7 | 23.3 | 0.0 | 3.3 | 6.7 | 0.0 | 0.0 |
| | Hard | 2.0 | 0.0 | 0.0 | 0.0 | 0.0 | 0.0 | 0.0 |
| DAG-First | Easy | 45.0 | 35.0 | 15.0 | 5.0 | 15.0 | 0.0 | 15.0 |
| | Medium | 10.0 | 20.0 | 3.3 | 6.7 | 10.0 | 0.0 | 6.7 |
| | Hard | 4.0 | 0.0 | 0.0 | 2.0 | 0.0 | 0.0 | 0.0 |

## B. Example Templates

This section shows examples of complete templates for each domain, including sample instantiations for each template. Graphical depictions shown in instantiations are for illustrative purposes only and are not provided to models.

**Math: Backtracking** Medium

**Template**

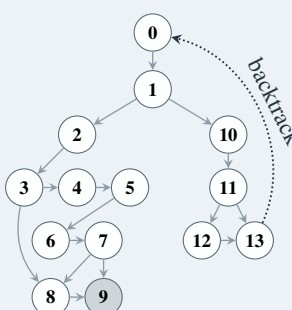

**Additional context:** This problem requires careful analysis of all preceding nodes.

---

**Instantiated Problem**

Solve this problem step by step and return the final solution at the end.

**Problem node_0:** The product of $N$ consecutive four-digit positive integers is divisible by [For this value use a number such that the sum of the prime factors of the answers to node_1, node_10, node_11, node_12, and node_13 is 2010]$^2$. What is the least possible value of $N$?

**Problem node_1:** What is the side length of the larger square if a small square is drawn inside a larger square, and the area of the shaded region and the area of the unshaded region are each [For this value use the answer from problem node_0 and add 13] cm$^2$?

**Problem node_2:** A factory is manufacturing solid aluminum cubes with a side length of [For this value use the answer from problem node_1 and add 4] mm. A chamfer process is then applied to the four edges on one face of each cube. The right-angled edge is chamfered that a sloping edge (45 degrees to each adjacent faces of the edge) with a width of $\sqrt{2}$ mm is formed. The cutted-off material is 100% recycled to make new cubes. How many chamfered cubes are needed to make the acuumulated recycled material enough for manufacturing another cube?

**Problem node_10:** Let $a, b$ be positive reals with $a > b > \frac{1}{2}a$. Place two squares of side lengths $a, b$ next to each other, such that the larger square has lower left corner at $(0, 0)$ and the smaller square has lower left corner at $(a, 0)$. Draw the line passing through $(0, a)$ and $(a + b, 0)$. The region in the two squares lying above the line has area [For this value use the answer from problem node_1 and add 2007]. If $(a, b)$ is the unique pair maximizing $a + b$, compute $\frac{a}{b}$.

**Problem node_3:** What is the 18 th digit after the decimal point of $\frac{\text{[For this value use the answer from problem node\_2 and add 9946]}}{9899}$?

**Problem node_11:** Find all prime numbers $p, q, r$, such that $\frac{p}{q} - \frac{\text{[For this value use the numerator of the reduced form of the fraction from problem node\_10 and subtract 1]}}{r+1} = 1$

**Problem node_4:** A sign has 31 spaces on a single line. The word RHOMBUS is written from left to right in [For this value use the answer from problem node_3 and add 2] consecutive spaces. There is an equal number of empty spaces on each side of the word. Counting from the left, in what space number should the letter $R$ be put?

**Problem node_12:** Let $ABC$ be a triangle with $\angle A =$ [For this value use the x-coordinate of the first ordered triple from problem node_11 and add 11]$^\circ$, $\angle B = 36^\circ$. Let $M$ be the midpoint of $AB$, $D$ a point on ray $CM$ such that $AB = AD$; $E$ a point on ray $BC$ such that $AB = BE$, and $F$ a point on ray $AC$ such that $AB = AF$. Find $\angle FDE$.

**Problem node_5:** How many ways are there to color every integer either red or blue such that $n$ and $n +$ [For this value use the answer from problem node_4 and subtract 6] are the same color for all integers $n$, and there does not exist an integer $k$ such that $k, k + 1$, and $2k$ are all the same color?

**Problem node_13:** Two mathematicians, Kelly and Jason, play a cooperative game. The computer selects some secret positive integer $n <$ [use the answer from problem node_12 and add use the x-coordinate of the first ordered

triple from problem node_11 and add 26] (both Kelly and Jason know that $n <$ [use the answer from problem node_12 and add use the x-coordinate of the first ordered triple from problem node_11 and add 26], but that they don't know what the value of $n$ is). The computer tells Kelly the unit digit of $n$, and it tells Jason the number of divisors of $n$. Then, Kelly and Jason have the following dialogue: Kelly: I don't know what $n$ is, and I'm sure that you don't know either. However, I know that $n$ is divisible by at least two different primes. Jason: Oh, then I know what the value of $n$ is. Kelly: Now I also know what $n$ is. Assuming that both Kelly and Jason speak truthfully and to the best of their knowledge, what are all the possible values of $n$?

**Problem node_6:** Let $\frac{1}{1-x-x^2-x^{\text{[For this value use the answer from problem node\_5 and subtract 3]}}} = \sum_{i=0}^{\infty} a_n x^n$, for what positive integers $n$ does $a\_n - 1 = n^2$?

**Problem node_7:** Find an ordered pair $(a, b)$ of real numbers for which $x^2 + ax + b$ has a non-real root whose cube is [For this value use the second integer in the answer list from problem node_6 and add 334].

**Problem node_8:** How many of the positive divisors of [use the x-coordinate from problem node_7 and add use the answer from problem node_3 and add 116] are perfect squares larger than 1?

**Problem node_9:** A rectangular pool table has vertices at $(0, 0)$([For this value use the x-coordinate from problem node_7 and add 5], 0)(0, [For this value use the answer from problem node_8 and add 7]), and ([For this value use the x-coordinate from problem node_7 and add 5], [For this value use the answer from problem node_8 and add 7]). There are pockets only in the four corners. A ball is hit from $(0, 0)$ along the line $y = x$ and bounces off several walls before eventually entering a pocket. Find the number of walls that the ball bounces off of before entering a pocket.

**What are the answers to problem node_9, node_3, node_0, and node_12?**

**Answer:** 9, 5, 5, 27

---

## CS: Compiler Easy

### Template

Here is an LLVM IR program.

```
[IR INSTANCE]
```

Definitions:
- Analyze only function @main.
- Instruction lines are non-empty, non-comment lines inside the function body that are not labels (lines ending with ':') or braces.
- Basic blocks are defined by label lines ending with ':', ordered by their appearance in the function body.
- A1, A2, ... refer to answers for earlier questions.
- Indices are 1-based for instruction lists, block lists, and pass lists.
- Line numbers are 1-based and refer to the full IR listing as shown.

Answer the following questions to the best of your ability.

**Q1: — QN:** (Concatenation of questions randomly selected for a pool of IR related questions).

### Instantiated Problem

Here is an LLVM IR program.

```
define i32 @helper(i32 %v) {
entry:
  %tmp = add i32 %v, 1
  ret i32 %tmp
}
```

```
define i32 @main(i32 %x, i32 %y) {
entry:
  %a = alloca i32, align 4
  %b = alloca i32, align 4
  store i32 %x, i32* %a, align 4
  store i32 %y, i32* %b, align 4
  %xval = load i32, i32* %a, align 4
  %yval = load i32, i32* %b, align 4
  %sum = add i32 %xval, %yval
  %dead = add i32 %sum, 7
  %cmp = icmp sgt i32 %sum, 10
  br i1 %cmp, label %then, label %else
then:
  %t1 = mul i32 %sum, 2
  br label %merge
else:
  %e1 = sub i32 %sum, 1
  br label %merge
merge:
  %phi = phi i32 [ %t1, %then ], [ %e1, %else ]
  %call = call i32 @helper(i32 %phi)
  %dead2 = add i32 %call, 42
  %ispos = icmp sgt i32 %call, 0
  br i1 false, label %unreachable, label %exit
unreachable:
  %u = add i32 %call, 1
  br label %exit
exit:
  ret i32 %call
}
```

Definitions:

- Analyze only function @main.
- Instruction lines are non-empty, non-comment lines inside the function body that are not labels (lines ending with ':') or braces.
- Basic blocks are defined by label lines ending with ':', ordered by their appearance in the function body.
- A1, A2, ... refer to answers for earlier questions.
- Indices are 1-based for instruction lists, block lists, and pass lists.
- Line numbers are 1-based and refer to the full IR listing as shown.

Answer the following questions to the best of your ability.

**Q1:** In function @main, how many 'br' instructions are there?

**Q2:** Let k = (A1 * 2). Which basic block index (by appearance, 1-based) contains the k-th instruction in @main?

**Q3:** Let k = (A2 * A2 + 8). What is the length (in characters) of the opcode of the k-th instruction in @main?

**Q4:** Let p be the pass at index (A3 - 3) in [dce, adce, simplifycfg, mem2reg]. Let k = (A3 * 3 + 9). Let op be the opcode of the k-th instruction in the original @main. After running 'opt -S -passes=p', how many times does op appear in that function?

**Q5:** Let k = (A4). In the original IR, how many instruction lines are in basic block k of @main (blocks ordered by appearance)?

**Q6:** Let p be the pass at index (A5 * 2 - 2) in [dce, adce, simplifycfg, mem2reg]. After running 'opt -S -passes=p', how many instruction lines are in function @main?

**Q7:** Let k = (A6 - 3). What is the 1-based position of the k-th instruction within its basic block in @main?

**Q8:** Let k = (A7 + 5). What is the 1-based line number of the label for basic block k in @main in the full original IR listing?

**Q9:** Let p be the pass at index (A8 - 31) in [dce, adce, simplifycfg, mem2reg]. How many instruction lines are removed from @main by running 'opt -S -passes=p'? (Removal = original count minus post-pass count.)

**Q10:** Let k = (A9 * 3 - 7). How many instruction lines appear in @main before basic block k begins?

**Answer:** 19

---

## CS: Matmul Hard

Template

Given the following chain of matrices and their sizes:

```
[MATRIX CHAIN INSTANCE]
```

Definitions:

- $M_i$ denotes the $i$-th matrix in the chain, with dimensions $r_i \times c_i$ as specified.
- A *parenthesization* specifies an order of evaluation for the product $M_1 * M_2 * \cdots * M_n$.
- The floating point operation count (Flops) for multiplying an $m \times k$ matrix by a $k \times n$ matrix is $2mkn$.
- Parenthesis *nesting depth* is the maximum number of simultaneously-open parentheses in the parenthesized expression.
- For Q4, after removing parentheses that are unnecessary for enforcing left-to-right evaluation of consecutive multiplications, the *distance* of a parenthesis pair is the number of matrices spanned by that pair.

Answer the following questions to the best of your ability.

**Q1:** What is the most efficient way to compute the product $M_1 * M_2 * \cdots * M_n$? Provide the answer as the expression with added parentheses. Provide the expression in the form of (M_x*M_y)... (no whitespace) and add *all* parentheses, even for trivial left-to-right chains.

**Q2:** What is the total number of floating point operations required for computing the multiplication chain obtained in Q1? (Remember: Flops for a matrix multiplication is $2mkn$.)

**Q3:** What is the deepest nesting level of parentheses in the decomposition obtained in Q1?

**Q4:** Assume consecutive multiplications in order from left to right do not need parentheses (e.g., (M_1*((M_2*M_3)*M_4)) becomes M_1*(M_2*M_3*M_4)). What is the longest distance between an opening and closing parenthesis (in number of matrices) in the decomposition obtained in Q1?

**Q5:** What is the difference between the answers of Q4 and Q3, multiplied by the answer for Q2?

**Provide your final answer in five lines, each starting with Q#: followed by the answer.**

---

Instantiated Problem

Given the following chain of matrices and their sizes:

```
M_1:  132x165
M_2:  165x1016
M_3:  1016x33
M_4:  33x93
M_5:  93x545
M_6:  545x1094
M_7:  1094x299
M_8:  299x232
M_9:  232x1286
M_10: 1286x1296
M_11: 1296x945
```

```
M_12: 945x963
M_13: 963x169
M_14: 169x291
M_15: 291x311
M_16: 311x131
M_17: 131x371
M_18: 371x678
M_19: 678x473
M_20: 473x1323
```

**Q1:** What is the most efficient way to compute the product M_1*M_2*...*M_20? Provide the answer as the expression with added parentheses. Provide the expression in the form of (M_x*M_y)... (no whitespace) and add *all* parentheses, even for trivial left-to-right chains.

**Q2:** What is the total number of floating point operations required for computing the multiplication chain obtained in Q1? Remember that the number of Flops for a matrix multiplication is $2mkn$.

**Q3:** What is the deepest nesting level of parentheses in the decomposition obtained in Q1?

**Q4:** Assume consecutive multiplications in order from left to right do not need parentheses (for example, (M_1*((M_2*M_3)*M_4)) becomes M_1*(M_2*M_3*M_4)), what is the longest distance between an opening and closing parentheses (in number of matrices) in the decomposition obtained in Q1?

**Q5:** What is the difference between the answers of Q4 and Q3, multiplied by the answer for Q2?

**Answer:**

```
Q1: ((M_1*(M_2*M_3))*(((((((((((((((M_4*M_5)*M_6)*M_7)*M_8)
*M_9)*M_10)*M_11)*M_12)*M_13)*M_14)*M_15)*M_16)*M_17)*M_18)*M_19)*M_20))
Q2: 468401208
Q3: 17
LR: M_1*(M_2*M_3)*(M_4*M_5*M_6*M_7*M_8*M_9*M_10*M_11*M_12
*M_13*M_14*M_15*M_16*M_17*M_18*M_19*M_20)
Q4: 17
LR: M_1*(M_2*M_3)*(M_4*M_5*M_6*M_7*M_8*M_9*M_10*M_11*M_12
*M_13*M_14*M_15*M_16*M_17*M_18*M_19*M_20)
Q5: 0
```

---

**Logic: Sokoban** Medium

Template

Sokoban is a puzzle game where you push boxes to goal positions. You are the player (@) and need to push all boxes ($) onto goal positions (.).

**Rules:**

1. You can move U (up), D (down), L (left), or R (right).

2. You can push boxes by moving into them (but only if there's space behind the box).

3. You cannot push boxes into walls (#) or other boxes.

4. You cannot pull boxes.

5. All boxes must be on goal positions to solve the puzzle.

**Symbols:**

- # = Wall (blocks movement)

- @ = Player

- $ = Box

- . = Goal position

- \* = Box on a goal

- + = Player on a goal

- (space) = Empty floor

You will be provided with a problem instance, given in the form:

- Grid: a list of lists where each inner list is a row of symbols

- Width, Height: dimensions of the grid

- Number of boxes and goals

- Player start location: (row, col)

- Box start locations: [(row, col), ...]

- Goal locations: [(row, col), ...]

**Puzzle instance:**

[PUZZLE INSTANCE]

Find a sequence of moves that pushes all boxes onto goal positions. Format your solution as:
```
solution = <string of moves>
```

---

Instantiated Problem

**Width:** 11
**Height:** 8
**Number of boxes and goals:** 5
**Player start:** (5, 2)
**Box locations:** [(2,3), (2,5), (3,2), (3,4), (3,6)]
**Goal locations:** [(5,5), (5,6), (5,7), (5,8), (5,9)]

```
[[ , #, #, #, #, #, #, #,  ,  ,  ],
 [#, #,  ,  ,  ,  ,  , #, #,  ,  ],
 [#,  ,  , $,  , $,  ,  , #,  ,  ],
 [#,  , $,  , $,  , $,  , #,  ,  ],
 [#, #,  , #, #, #,  , #, #, #, #],
 [ , #, @,  ,  , ., ., ., ., ., #],
 [ , #, #,  ,  ,  ,  ,  , #, #, #],
 [ ,  , #, #, #, #, #, #, #,  ,  ]]
```

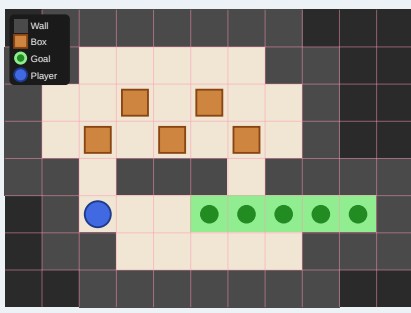

**Answer:**
*Note: This puzzle has multiple valid solutions. One possible solution is:*

```
solution = UURRUURRDDDUUUULLDDLLDDRRRRRRLLUUUULLDDRL
UURRDDDULLLLLDDRRRRRLUUUULLDRURDDDULLLLLDDRDRRRRULL
RUUUULLLLLDRRRURDDDUULLLLLLDRDDRDRRURUUULLLLLDRRRURD
```

```
DULLLLLDDRR
```

(144 moves)

---

## Logic: Sudoku Hard

Template

A Sudoku puzzle is a grid where each row, column, and box must contain all digits from $1$ to $n$ exactly once. Some cells are pre-filled with numbers (clues), and you need to fill in the remaining empty cells.
**Rules:**

1. Each row must contain all digits $1$–$n$ exactly once.

2. Each column must contain all digits $1$–$n$ exactly once.

3. Each of the $n$ boxes (non-overlapping subgrids) must contain all digits $1$–$n$ exactly once.

4. Only use digits $1$–$n$ (empty cells are represented as $0$ in the input).

You will be provided with a problem instance, given in the form:

- Grid size: side $\times$ side (where side $=$ block_size $\times$ block_size)

- Block size: block_size $\times$ block_size

- Puzzle grid: $[\text{row}_0, \text{row}_1, \ldots, \text{row}_{n-1}]$

**Puzzle instance:**

[PUZZLE INSTANCE]

Find the complete solution to this Sudoku puzzle. Format your solution as:
```
solution = [row_0, row_1, ..., row_{n-1}]
```

---

Instantiated Problem

**Grid size:** $9 \times 9$
**Block size:** $3 \times 3$
**Puzzle grid:**

```
[[0, 2, 0, 0, 0, 0, 0, 0, 9],
 [0, 5, 0, 1, 0, 0, 2, 0, 0],
 [6, 0, 0, 7, 0, 2, 0, 5, 0],
 [0, 0, 0, 0, 1, 0, 0, 0, 0],
 [0, 0, 6, 8, 0, 5, 0, 2, 0],
 [0, 0, 0, 0, 0, 0, 0, 0, 3],
 [0, 6, 0, 2, 0, 8, 0, 7, 0],
 [7, 0, 0, 0, 6, 0, 4, 0, 0],
 [8, 0, 0, 0, 0, 0, 0, 0, 0]]
```

| | 2 | | | | | | | 9 |
|---|---|---|---|---|---|---|---|---|
| | 5 | | 1 | | | 2 | | |
| 6 | | | 7 | | 2 | | 5 | |
| | | | | 1 | | | | |
| | | 6 | 8 | | 5 | | 2 | |
| | | | | | | | | 3 |
| | 6 | | 2 | | 8 | | 7 | |
| 7 | | | | 6 | | 4 | | |
| 8 | | | | | | | | |

**Answer:**

```
solution =
[[1, 2, 3, 4, 5, 6, 7, 8, 9],
 [4, 5, 7, 1, 8, 9, 2, 3, 6],
 [6, 8, 9, 7, 3, 2, 1, 5, 4],
 [2, 7, 8, 9, 1, 3, 6, 4, 5],
 [3, 1, 6, 8, 4, 5, 9, 2, 7],
 [9, 4, 5, 6, 2, 7, 8, 1, 3],
 [5, 6, 4, 2, 9, 8, 3, 7, 1],
 [7, 3, 2, 5, 6, 1, 4, 9, 8],
 [8, 9, 1, 3, 7, 4, 5, 6, 2]]
```

| 1 | 2 | 3 | 4 | 5 | 6 | 7 | 8 | 9 |
|---|---|---|---|---|---|---|---|---|
| 4 | 5 | 7 | 1 | 8 | 9 | 2 | 3 | 6 |
| 6 | 8 | 9 | 7 | 3 | 2 | 1 | 5 | 4 |
| 2 | 7 | 8 | 9 | 1 | 3 | 6 | 4 | 5 |
| 3 | 1 | 6 | 8 | 4 | 5 | 9 | 2 | 7 |
| 9 | 4 | 5 | 6 | 2 | 7 | 8 | 1 | 3 |
| 5 | 6 | 4 | 2 | 9 | 8 | 3 | 7 | 1 |
| 7 | 3 | 2 | 5 | 6 | 1 | 4 | 9 | 8 |
| 8 | 9 | 1 | 3 | 7 | 4 | 5 | 6 | 2 |

## Chemistry: Reactions and Molecular Understanding Medium

Template

**Subproblem 1:** Given the following set of molecules (SMILES): Molecule A: A Molecule B: `B` and Molecule `C`: `C`, select explicitly the molecule which would have the greatest number of unique (non-hashed) radius 2 morgan fingerprint bits. Consider the selected mol as *Mol-1*.

**Subproblem 2:** Choose [SMILES] from the following options (`2.1`, `2.2`, `2.3`) that is equivalent to the following SMILES `Target-2`. Consider the selected mol as *Mol-2*.

**Subproblem 3:** Using the molecules selected in Subproblem #1 (*Mol 1*) and Subproblem #2 (*Mol 2*), predict the major product (SMILES) formed after performing a reaction under the given conditions: Solvent: C(Cl)(Cl)Cl and Temperature: 0.0°C. If any essential reaction components or conditions are not specified, use reasonable standard choices based on the reaction type. Consider the selected mol as *Mol 3*.

**Subproblem 4:** Select among the following SMILES strings (`4.1`, `4.2`, `4.3`), the molecule matching the empirical formula [('C', 4), ('H', 11), ('N', 1), ('O', 1)]. Consider the selected mol as *Mol-4*.

**Subproblem 5:** Using the reaction product obtained in Subproblem #3 (*Mol 3*) and the molecule selected in Subproblem #4 (*Mol 4*), predict the major product (SMILES) formed after performing a reaction under the given conditions: Solvent: CN(C)C=O and Temperature: 90°C. If any essential reaction components or conditions are not specified, use reasonable standard choices based on the reaction type. Consider the selected mol as Mol 5.

**Subproblem 6:** Select among the following SMILES strings (`6.1`, `6.2`, `6.3`) the molecule with the greatest number of implicit hydrogens. Consider the selected mol as *Mol-6*.

**Subproblem 7:** Select among the following SMILES strings (`7.1`, `7.2`, `7.3`) the molecule with the 2nd greatest number of implicit hydrogens. Consider the selected mol as *Mol-7*.

**Subproblem 8:** Using the molecules selected in Subproblem #6 (*Mol 6*) and Subproblem #7 (*Mol 7*), predict the major product (SMILES) formed after performing a reaction under the given conditions: Solvent: C(Cl)(Cl)Cl and Temperature: 65°C. If no essential reaction components or conditions are specified, use reasonable standard choices based on the reaction type. Consider the selected mol as *Mol 8*.

**Subproblem 9:** Consider *Mol-5*, *Mol-7* and *Mol-8*. Provide the number of bonds in the path that is associated with the topological diameter of each molecule. The topological distance between two atoms is the shortest path length (in bonds) connecting them. The topological diameter is the largest of these shortest path lengths, i.e. the maximum graph distance between any two atoms. Format this as a list for the molecules in the order they are given, for example [5, 12, 9]. When you are done, use the following format for your final answer and the molecules found along the way - Final Answer: XXX Mol-1: XXX Mol-2: XXX ..."

Instantiated Problem

**Subproblem 1:** Given the following set of molecules (SMILES): Molecule A: ClC=1C=C(C(=O)OO)C=CC1, Molecule B: CC(C)Sc1ccccc1 and Molecule C: CC(C)(C)N1CCNCC1=O, select explicitly the molecule which would have the greatest number of unique (non-hashed) radius 2 morgan fingerprint bits. Consider the selected mol as *Mol-1*.

**Subproblem 2:** Choose the [SMILES] from the following options (c1(C(=O)OCC)ncc2nc(C(C)(C)C)sc2c1O, c1(C(=O)N)sc2nc(SC)nc(-c3cc(ccn3)OC)c2c1N, c1cc(OC2CCCCC2)cc(c1C(=O)O)Nc1ccc(F)cc1) that is equivalent to the following SMILES NC1=C(SC=2N=C(N=C(C21)C2=NC=CC(=C2)OC)SC)C(=O)N. Consider the selected mol as *Mol-2*.

**Subproblem 3:** Using the molecules selected in Subproblem #1 (*Mol 1*) and Subproblem #2 (*Mol 2*), predict the major product (SMILES) formed after performing a reaction under the given conditions: Solvent: C(Cl)(Cl)Cl and Temperature: 0.0°C. If any essential reaction components or conditions are not specified, use reasonable standard choices based on the reaction type. Consider the selected mol as *Mol 3*.

**Subproblem 4:** Select among the following SMILES strings (NC(CO)(C)C, [O]=[Mn](=[O])([O-])[O-], C=C(CCl)CCl), the molecule matching the empirical formula [('C', 4), ('H', 11), ('N', 1), ('O', 1)]. Consider the selected mol as *Mol-4*.

**Subproblem 5:** Using the reaction product obtained in Subproblem #3 (Mol 3) and the molecule selected in Subproblem #4 (Mol 4), predict the major product (SMILES) formed after performing a reaction under the given conditions: Solvent: CN(C)C=O and Temperature: 90°C. If any essential reaction components or conditions are not specified, use reasonable standard choices based on the reaction type. Consider the selected mol as Mol 5.

**Subproblem 6:** Select among the following SMILES strings (ClN1C(CCC1=O)=O, Clc1ccc(Cl)nn1, O=Cc1cocn1) the molecule with the largest number of implicit hydrogens. Consider the selected mol as *Mol-6*.

**Subproblem 7:** Select among the following SMILES strings ([N-]=[N+]=Nc1c(Cl)cncc1C=O, CC1(C)[C@H](C=C(Cl)Cl)[C@@H]1C(=O)O, FC=1C=CC=2N(C1)C=NN2) the molecule with the 2nd greatest number of implicit hydrogens. Consider the selected mol as *Mol-7*.

**Subproblem 8:** Using the molecules selected in Subproblem #6 (Mol 6) and Subproblem #7 (*Mol 7*), predict the major product (SMILES) formed after performing a reaction under the given conditions: Solvent: C(Cl)(Cl)Cl and Temperature: 65°C. If no essential reaction components or conditions are specified, use reasonable standard choices based on the reaction type. Consider the selected mol as Mol 8.

**Subproblem 9:** Consider *Mol-5*, *Mol-7* and *Mol-8*. Provide the number of bonds in the path that is associated with the topological diameter of each molecule. The topological distance between two atoms is the shortest path length (in bonds) connecting them. The topological diameter is the largest of these shortest path lengths, i.e., the maximum graph distance between any two atoms. Format this as a list for the molecules in the order they are given, for example [5, 12, 9]. When you are done, use the following format for your final answer and the molecules found along the way - Final Answer: XXX Mol-1: XXX Mol-2: XXX ..."

**Answer: [11, 5, 5]**

**Molecular structures of SMILES in above questions for illustration.**

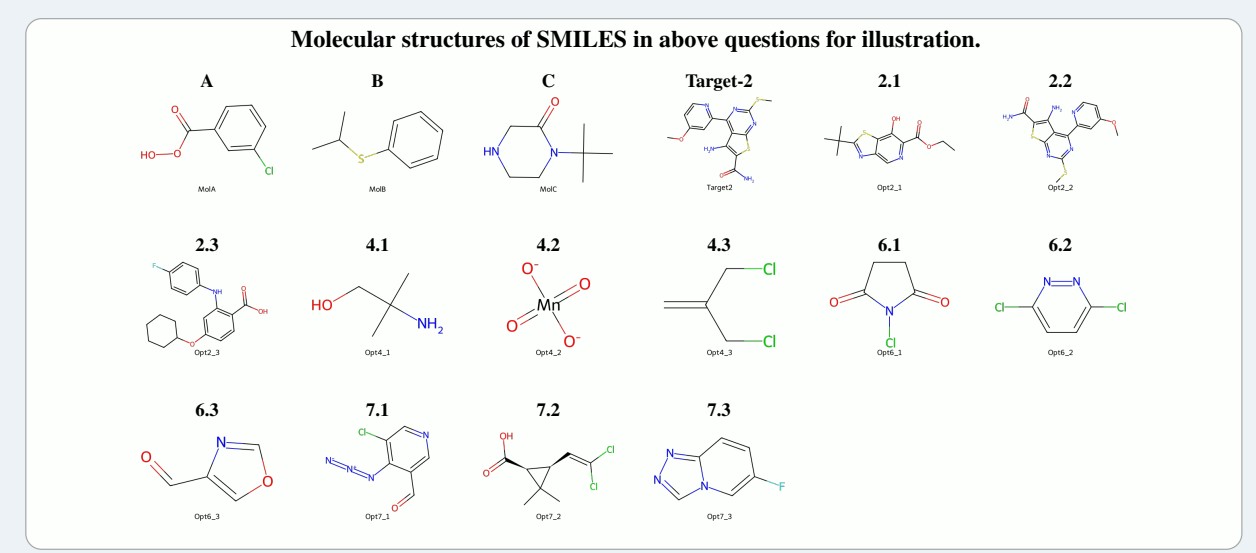

## Chemistry: Protein Structure Medium

Template

Please use the following ATOM record to answer the question composed from six subproblems. Each subproblem builds on one another where the question answer is the solution to the final subproblem.

[ATOM RECORD]

**Subproblem 1:** Given the ATOM record: Identify the positions of all missing residue names in the protein sequence. Consider the span of missing residues as *Gap-1*.

**Subproblem 2:** Given the ATOM record and using the residue positions selected in Subproblem #1 (*Gap-1*), measure the compactness of the protein backbone before and after *Gap-1*. Compactness is defined as the Euclidean distance between non-consecutive alpha carbon atoms: CA(i) - CA(i+3) for a residue position i. Give your answer in Angstroms. Consider the compactness measure as *Feature-1*.

**Subproblem 3:** Given the ATOM record and using the residue positions selected in Subproblem #1 (*Gap-1*), measure the average backbone dihedral angles for the residues before and after *Gap-1*. For both the residues surrounding the gap return the angles phi ($\phi$) and psi ($\psi$) measured in degrees. Phi is the angle defined by atoms in the sequence: C(i-1)-N(i)-CA(i)-C(i). Psi is the angle defined by atoms in the sequence: N(i)-CA(i)-C(i)-N(i+1). Consider these two pairs of angles as *Feature-2*.

**Subproblem 4:** Using the compactness measure *Feature-1* from Subproblem #2 and the dihedral angles *Feature-2* from Subproblem #3, predict the secondary structure of the missing residue *Gap-1*. Consider the secondary structure prediction as *Struct-1*.

**Subproblem 5:** Using the secondary structure prediction obtained in Subproblem #4 (*Struct-1*) and the patterns in a window of [WINDOW SIZE] residues around *Gap-1*, Determine the identity of the missing residues in *Gap-1*. Consider the predicted missing residues as *Res-1*

**Subproblem 6:** Using the secondary structure prediction obtained in Subproblem #4 (*Struct-1*) and the missing residues, *Res-1*, rank the residues in sequence positions [RANDOM RESIDUES] and *Gap-1* according to the following criteria:

Criterion 1: If the predicted structure is an $\alpha$-helix order the residues in descending order of neighbor count.

Criterion 2: If the predicted structure is a $\beta$-strand or loop region, order the residues in descending order of Euclidean distance from each residue CA atom to the protein geometric center.

Where neighbor count is defined for each residue as the number of CA atoms within 12 Å of CA(i) (excluding i-2 to i+2). Where the protein geometric center is the point defined by the mean of all ATOM record CA x-coordinates, y-coordinates, and z-coordinates. If there is a tie, order elements by their residue position in the sequence. Provide the list of residue names ranked by the appropriate criteria as Result: [ANSWER]

---

Instantiated Problem

Please use the following ATOM record to answer the question composed from six subproblems. Each subproblem builds on one another where the question answer is the solution to the final subproblem.

| Record Type | Atom Number | Atom Name | Residue Name | Residue Number | X | Y | Z | Occupancy | Temperature Factor | Element Name |
|---|---|---|---|---|---|---|---|---|---|---|
| | | | | | ... | | | | | |
| ATOM | 238 | N | ILE | 34 | 11.49 | 15.773 | 7.038 | 1 | 5.52 | N |
| ATOM | 239 | CA | ILE | 34 | 12.552 | 15.877 | 6.036 | 1 | 6.82 | C |
| ATOM | 240 | C | ILE | 34 | 13.59 | 16.917 | 6.56 | 1 | 6.92 | C |
| ATOM | 241 | O | ILE | 34 | 13.168 | 18.006 | 6.945 | 1 | 9.22 | O |
| ATOM | 242 | CB | ILE | 34 | 11.987 | 16.36 | 4.681 | 1 | 8.11 | C |
| ATOM | 243 | CG1 | ILE | 34 | 10.914 | 15.338 | 4.163 | 1 | 9.59 | C |
| ATOM | 244 | CG2 | ILE | 34 | 13.131 | 16.517 | 3.629 | 1 | 9.73 | C |
| ATOM | 245 | CD1 | ILE | 34 | 10.151 | 16.024 | 2.938 | 1 | 13.41 | C |
| ATOM | | | | | 15.592 | 16.974 | 9.434 | 1 | 9.46 | |
| ATOM | | | | | 16.622 | 16.995 | 8.285 | 1 | 8.07 | |
| ATOM | | | | | 18.298 | 15.206 | 9.219 | 1 | 9.85 | |
| ATOM | | | | | 17.36 | 15.651 | 8.067 | 1 | 9.41 | |
| ATOM | | | | | 17.097 | 16.66 | 4.97 | 1 | 7.9 | O |
| ATOM | | | | | 14.856 | 16.493 | 6.536 | 1 | 7.06 | N |
| ATOM | | | | | 16.913 | 17.55 | 5.819 | 1 | 6.63 | C |
| ATOM | | | | | 15.93 | 17.454 | 6.941 | 1 | 7.52 | C |
| ATOM | | | | | 20.593 | 17.742 | 3.945 | 1 | 9.09 | O |
| ATOM | | | | | 18.945 | 20.364 | 4.783 | 1 | 9.67 | |
| ATOM | | | | | 17.664 | 18.669 | 5.806 | 1 | 8.07 | N |
| ATOM | | | | | 17.371 | 19.9 | 6.596 | 1 | 9.53 | |
| ATOM | | | | | 19.925 | 18.042 | 4.949 | 1 | 8.31 | C |
| ATOM | | | | | 18.238 | 20.937 | 5.908 | 1 | 10.15 | |
| ATOM | | | | | 18.635 | 18.861 | 4.738 | 1 | 8.78 | C |
| ATOM | 261 | N | GLY | 37 | 20.172 | 17.73 | 6.217 | 1 | 8.48 | N |
| ATOM | 262 | CA | GLY | 37 | 21.452 | 16.969 | 6.513 | 1 | 9.2 | C |
| ATOM | 263 | C | GLY | 37 | 21.143 | 15.478 | 6.427 | 1 | 10.41 | C |
| ATOM | 264 | O | GLY | 37 | 20.138 | 15.023 | 5.878 | 1 | 12.06 | O |
| | | | | | ... | | | | | |

**Subproblem 1:** Given the ATOM record: Identify the positions of all missing residue names in the protein sequence. Consider the span of missing residues as *Gap-1*.

**Subproblem 2:** Given the ATOM record and using the residue positions selected in Subproblem #1 (*Gap-1*), measure the compactness of the protein backbone before and after *Gap-1*. Compactness is defined as the Euclidean distance between non-consecutive alpha carbon atoms: CA(i) - CA(i+3) for a residue position i. Give your answer in Angstroms. Consider the compactness measure as *Feature-1*.

**Subproblem 3:** Given the ATOM record and using the residue positions selected in Subproblem #1 (*Gap-1*), measure the average backbone dihedral angles for the residues before and after *Gap-1*. For both the residues surrounding the gap return the angles phi ($\phi$) and psi ($\psi$) measured in degrees. Phi is the angle defined by atoms in the sequence: C(i-1)-N(i)-CA(i)-C(i). Psi is the angle defined by atoms in the sequence: N(i)-CA(i)-C(i)-N(i+1). Consider these two pairs of angles as *Feature-2*.

**Subproblem 4:** Using the compactness measure *Feature-1* from Subproblem #2 and the dihedral angles *Feature-2* from Subproblem #3, predict the secondary structure of the missing residue *Gap-1*. Consider the secondary structure prediction as *Struct-1*.

**Subproblem 5:** Using the secondary structure prediction obtained in Subproblem #4 (*Struct-1*) and the patterns in a window of three residues around *Gap-1*, Determine the identity of the missing residues in *Gap-1*. Consider the predicted missing residues as *Res-1*

**Subproblem 6:** Using the secondary structure prediction obtained in Subproblem #4 (*Struct-1*) and the missing residues, *Res-1*, rank the residues in sequence positions [23, 31, 41, 44, 46] and *Gap-1* according to the following criteria:

  Criterion 1: If the predicted structure is an $\alpha$-helix order the residues in descending order of neighbor count.

  Criterion 2: If the predicted structure is a $\beta$-strand or loop region, order the residues in descending order of Euclidean distance from each residue CA atom to the protein geometric center.

Where neighbor count is defined for each residue as the number of CA atoms within 12 Å of CA(i) (excluding i-2 to i+2). Where the protein geometric center is the point defined by the mean of all ATOM record CA x-coordinates, y-coordinates, and z-coordinates. If there is a tie, order elements by their residue position in the sequence. Provide the list of residue names ranked by the appropriate criteria as Result: [ANSWER]

**Answer:** [PRO, PRO, ILE, TYR, ASN, GLU, GLY]

---

**Chess: Minimax Pawn Capture** Hard

Template

There is a 30 x 30 chessboard with one knight and some pawns on it. You are given two integers kx and ky where (kx, ky) denotes the position of the knight, and a 2D array positions where positions[i] = [xi, yi] denotes the position of the pawns on the chessboard.

Alice and Bob play a turn-based game, where Alice goes first.

In each player's turn: The player selects a pawn that still exists on the board and captures it with the knight in the fewest possible moves. Note that the player can select any pawn, it might not be one that can be captured in the least number of moves.

In the process of capturing the selected pawn, the knight may pass other pawns without capturing them. Only the selected pawn can be captured in this turn.

Alice is trying to maximize the sum of the number of moves made by both players until there are no more pawns on the board, whereas Bob tries to minimize them.

Return the maximum total number of moves made during the game that Alice can achieve, assuming both players play optimally.

Note that in one move, a chess knight has eight possible positions it can move to. Each move is two cells in a cardinal direction, then one cell in an orthogonal direction.

Now, solve the problem for the following setup:
Board size: [BOARD DIMENSIONS]
KX, KY (knight position): [KNIGHT POSITION]
Pawn positions: [PAWN POSITIONS]
Your response should be in the format: FINAL_ANSWER: maximum total number of moves
There should be NO ADDITIONAL OUTPUT after the final answer.

There is a 30 x 30 chessboard with one knight and some pawns on it. You are given two integers kx and ky where (kx, ky) denotes the position of the knight, and a 2D array positions where positions[i] = [xi, yi] denotes the position of the pawns on the chessboard.
Alice and Bob play a turn-based game, where Alice goes first.
In each player's turn: The player selects a pawn that still exists on the board and captures it with the knight in the fewest possible moves. Note that the player can select any pawn, it might not be one that can be captured in the least number of moves.
In the process of capturing the selected pawn, the knight may pass other pawns without capturing them. Only the selected pawn can be captured in this turn.
Alice is trying to maximize the sum of the number of moves made by both players until there are no more pawns on the board, whereas Bob tries to minimize them.
Return the maximum total number of moves made during the game that Alice can achieve, assuming both players play optimally.
Note that in one move, a chess knight has eight possible positions it can move to. Each move is two cells in a cardinal direction, then one cell in an orthogonal direction.

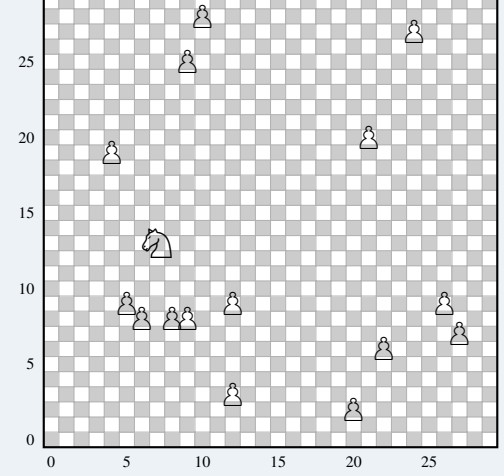

Now, solve the problem for the following setup:
Board size: 30 x 30
KX, KY (knight position): [7, 13]
Pawn positions: [[21, 20], [12, 3], [24, 27], [5, 9], [27, 7], [22, 6], [12, 9], [9, 8], [9, 25], [4, 19], [20, 2], [8, 8], [26, 9], [10, 28], [6, 8]]
Your response should be in the format: FINAL_ANSWER: maximum total number of moves
There should be NO ADDITIONAL OUTPUT after the final answer.

**Answer:** 106

**Give the next best move in response to this FEN puzzle.**
Think thoroughly until you get the optimal solution. After your reasoning, return only the next best move and nothing else.
FEN: [FEN]

Instantiated Problem

**Give the next best move in response to this FEN puzzle.**

Think thoroughly until you get the optimal solution. After your reasoning, return only the next best move and nothing else.

FEN: 2r3k1/p2R1nBp/2r3p1/q2N1p2/1bQ2P2/1P2P3/2R2K2/8 w - - 0 1

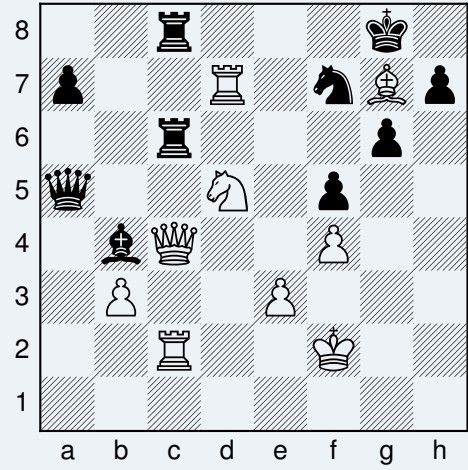

**Answer:** Qd4

## C. Additional Analysis

**LongCoT-mini.** LongCoT-mini (500 specifically generated easy questions) allows us to better analyze the performance of open-source models that achieve near-zero performance on LongCoT. We present a domain wise distribution of performance in Figure 9. Overall, GPT 5.2 does significantly better than all other models. As discussed in Section 4.2, the parameterized structure of our benchmark allows us to study how performance degrades across horizon lengths with finer granularity and isolation than short reasoning benchmarks or agentic tasks typically offer.

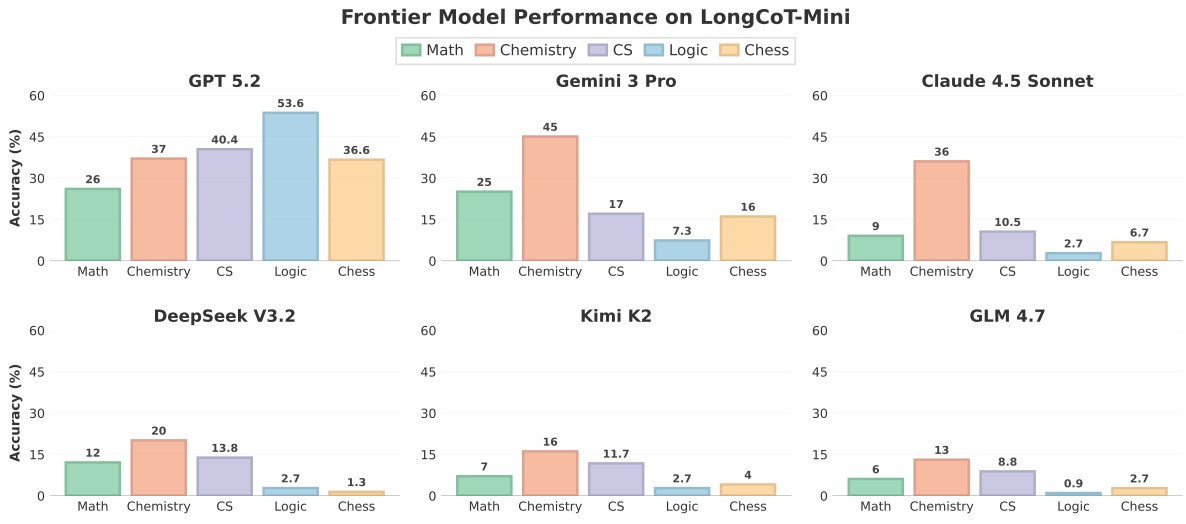

*Figure 9.* LongCoT-mini domain-specific results provide more signal on model performance. These findings comport with the design goals of LongCoT: rather than deep domain knowledge, LongCoT success demands the ability to reason over long horizons.

**Long-Horizon Reasoning Challenges.** In general, we observe the following fundamental issues in long-horizon reasoning capabilities. Poor early planning commits models to inefficient strategies, and errors compound across steps as single incorrect intermediate values propagate through dependent subproblems. Models also exhaust their effective context and resort to guessing on later steps, or give up prematurely on problems they can sometimes solve when given a fresh attempt. Finally, models often fail to backtrack and explore valid alternative paths. These errors are not independent of each other. We observe specific behaviors in our analysis of the reasoning traces of open-source frontier models:

- Models often make inefficient or incorrect plans early on, especially for our implicit procedural domains. After a certain point in the reasoning chain, they recognize this but do not backtrack and instead continue down an incorrect path.

- Models make compounding errors across steps and are unable to detect these, especially when the errors occurred early on and not immediately prior.

- Models give up very early in their reasoning chains, sometimes even when independent sampling would yield trajectories that solve the task.

- Models often skip important steps or guess intermediate values to simplify reasoning, which often causes incorrect downstream answers.

- Sometimes models cannot reason over the right sequence of future steps and exhaust their output context due to an inefficient plan. This often results in rushed or guessed final answers as models near their token limit, even when a more efficient reasoning path existed from the start.

- Models attempt shortcuts to reasoning problems, which causes looping and step-level execution failures that lead to incorrect answers. Rather than systematically working through the problem structure, models try to pattern-match or jump to conclusions, and when this fails they often repeat similar failing approaches rather than switching strategies.

We observed these behaviors consistently and believe LongCoT will serve as a milestone for improving these capabilities, advancing research on long-horizon reasoning and yielding practical benefits for complex autonomous tasks.

**Compositional Reasoning vs. Context Length** A natural question is whether the difficulty of our benchmark arises from compositional dependencies between subproblems or simply from producing long outputs. To test this, we run an ablation over a subset of LongCoT-Math questions where the same subproblems are presented within a single prompt but with no inter-node dependencies. The model answers all questions independently in a single output and is scored on the same leaf nodes as the composed setting. We present these results in Figure 10.

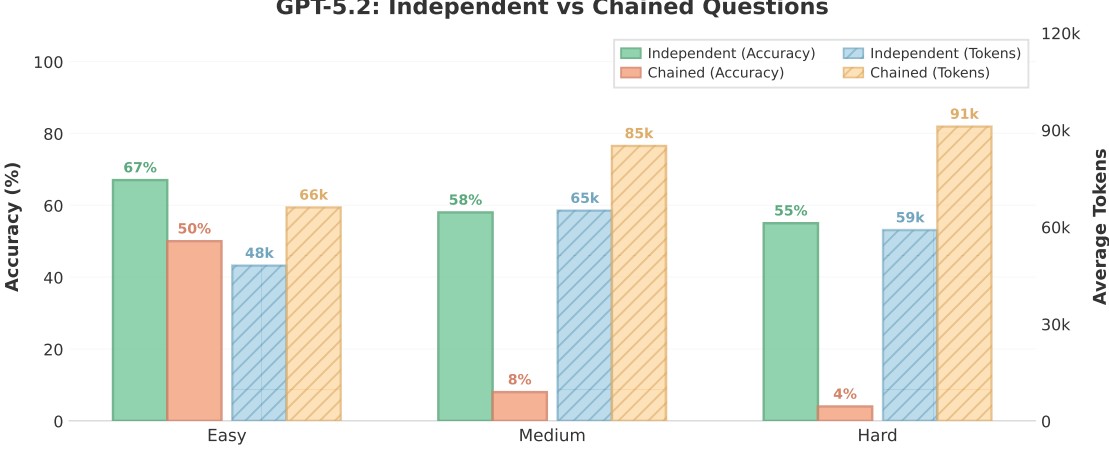

*Figure 10.* GPT-5.2 accuracy and token usage on independent versus composed LongCoT-Math questions. Independent questions are presented in a single prompt with no inter-node dependencies and scored on the same leaf nodes. Accuracy drops sharply when questions are composed while token usage remains comparable, confirming that compositional dependency, not just output length, drives difficulty.

If difficulty were a function of independent questions combined with a long output, we would expect accuracy to be similar under both conditions. Instead, on medium-difficulty questions, GPT-5.2 scores 58% on independent questions versus 8% when composed; on hard questions, 55% versus 4% (Figure 9). Token usage is comparable across both conditions,

confirming that compositional dependency structure, not context length, is the primary driver of failure. We also note that our prompts are approximately 2k tokens and the extended context is entirely model-generated, so any context-length effects reflect the model's own reasoning process rather than an externally imposed memory challenge.

**Recursive Language Models.**    Recursive Language Models (RLMs) are a very recent framework where language models recursively call sub-agents to decompose problems and manage context (Zhang et al., 2025). The root model interacts with a REPL environment storing the context as a variable, and can spawn recursive calls to partition, summarize, or query over subsets of this context. RLMs have shown incredible results on long-context benchmarks like BrowseComp (Wei et al., 2025), where context decomposition and iterative refinement try to mitigate context degradation.

While LongCoT is designed to measure CoT reasoning abilities directly, we test its resistance to scaffolding approaches. We evaluate a random subset of questions with GPT-5.2 RLM in two settings: a reasoning-only configuration where sub-agents solve problems without executing code (configured with appropriate prompts), aligned with LongCoT's goal of isolating reasoning capability, and the default RLM configuration where sub-agents may write and execute simulation code (with chess, rdkit, and other domain libraries available). In the reasoning-only setting, RLM with GPT 5.2 does not outperform GPT 5.2 alone. LongCoT's graph-structured dependencies make problem decomposition harder, and context between sub-agent calls sometimes loses useful information (such as failed exploration paths or variable dependencies) that later reasoning steps require. Zhang et al. (2025) show how skills such as backtracking effectively or better decomposing tasks can be improved with specific prompting, thus improving performance on LongCoT.

Even when code simulation is enabled, performance improves primarily on implicit domains where substantial parts of the dependency structure can be externalised to code (Logic: 68.3%, Chess: 30.6%, CS: 26.7%), while explicit compositional domains (Math, Chemistry) remain more challenging. Thus, code execution helps on some implicit tasks, but does not remove the core difficulty of LongCoT. The goal of our benchmark remains measuring CoT reasoning abilities directly, and these results confirm that strong performance requires long-horizon reasoning within a well-structured chain of thought.

We analyze specific parts of LongCoT that are challenging for RLMs. On explicit compositional problems, RLMs face decomposition challenges: subagents either receive all subproblems at once, or receive isolated subproblems without the dependency information or context needed to aggregate answers correctly. The iterative problem solving process can also sometimes compound this. After each subagent call, the root model proceeds to the next step, but later steps often need more context and variables than are being actively tracked, and unsuccessful exploration paths are not explicitly carried between iterations, so subagents may revisit approaches that have already proven unproductive. On implicit procedural problems, planning is sometimes the limiting factor: the root model cannot always anticipate the branching structure required for problems like minimax games or constraint satisfaction, so the full search space is not always explored. These issues illustrate that tasks with graph-structured dependencies during the reasoning process can benefit from structured approaches that target certain capabilities, especially with prompt-tuning or training.

**Reasoning Trace Analysis**    To characterize the structure of extended reasoning traces, we segmented each trace into 100 equal-length chunks and classified each chunk into one of six categories—Setup, Planning, Solving, Verification, Stuck, and Backtracking—using GPT-5-mini as an annotator. Because closed-source frontier models do not expose reasoning traces, we restrict this analysis to open-source models. Success rates for these models on the full benchmark are below 2%, providing too few correct traces for meaningful comparison. We therefore draw traces from LongCoT-mini problems, where success rates are high enough to compare correct and incorrect reasoning behavior on the same templates. We analyzed 20 correct and 20 incorrect traces for DeepSeek V3.2, and 16 correct and 16 incorrect for Kimi K2 Thinking. Across both models, incorrect traces exhibited roughly twice the rate of backtracking compared to correct traces (DeepSeek: 4.2% vs. 2.3%; Kimi: 4.2% vs. 2.0%), while correct traces allocated more reasoning budget to setup and problem comprehension (DeepSeek: 20.6% vs. 17.0%; Kimi: 21.6% vs. 17.9%). In the setup phase, models restate constraints, initial state, identify the goal state, and parse the problem. The dominant category in all cases was direct solving (62–68%), with the remaining budget split among verification, planning, and error-recovery behaviors. A simple MLP trained on the six-dimensional category count vector achieves around 75% test accuracy at distinguishing correct from incorrect traces, suggesting that coarse behavioral patterns could carry a detectable but modest signal about outcome correctness. We show a subset of these categorized reasoning traces in Section 4.3 and Figure 11. While we provide a very limited analysis of the structure of reasoning traces, this could be an interesting avenue for future work.

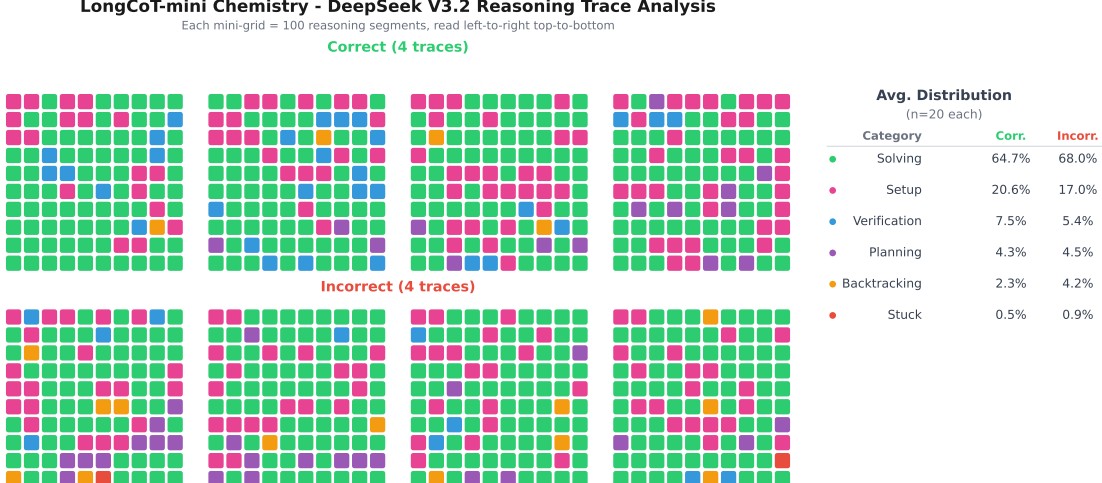

*Figure 11.* Reasoning trace structure for DeepSeek V3.2 on LongCoT-mini Chemistry Problems. Each trace is segmented into a 10×10 grid (read *left-to-right, top-to-bottom*), classified into Setup, Planning, Solving, Verification, Stuck, or Backtracking. Correct traces (top) allocate more budget to setup, while incorrect traces (bottom) show visibly more backtracking (orange) and stuck (red) segments.

## D. Limitations and Future Work

Our goal with LongCoT is to inspire future work on environments, evaluations, and training techniques (Motwani et al., 2025; Aghajohari et al., 2025) that improve long-horizon reasoning capabilities. Additionally, extending benchmarking of long-horizon reasoning capabilities to cover more domains could help ensure that long-horizon abilities improve independent of their context. We note that the very high cost of evaluations (often over $1 USD per API call) prevents our reporting of statistics like pass@k and majority voting scores. Nonetheless, the size of our benchmark is large enough to provide a rigorous analysis. We faced several issues with GPT 5.2 endpoints providing different accuracies/timeouts through different providers we had budgets for, and the next version of this work will reconduct GPT 5.2 evals on Mathematics and Computer Science directly with the OpenAI API for greater consistency.

We also do not train models specifically for these tasks, which future work could explore (see for e.g. RLVE (Zeng et al., 2025) which procedurally generates adaptive verifiable environments). Notably, compute for such training would be substantial, as models regularly output nearly 100K tokens on LongCoT questions. This suggests future work could also (more tractably) focus on training-free solutions like RLMs (Zhang et al., 2025), which we included a preliminary study of. Overall, LongCoT is intended as a reasoning-only benchmark, and extension to other settings remains part of future work.

Finally, while we do our best to keep the tasks in LongCoT as realistic as possible (given the domain), not all questions are necessarily directly applicable to realistic workflows. However, for each question in LongCoT, success depends on long horizon reasoning capabilities, which in turn are critical for real-world economically-valuable agentic tasks. Moreover, LongCoT isolates these capabilities, testing them independently of tests of scaffolding and tool-calling, which agentic benchmarks conflate with tests of long-horizon reasoning. We release LongCoT as a public benchmark.

