# OpenReview forum: "LongCoT: Benchmarking Long-Horizon Chain-of-Thought Reasoning"
_ICML.cc/2026/Conference — ICML 2026 regular_

### Official Review · Reviewer_qHXU · 2026-03-06

**Soundness:** 2
**Presentation:** 3
**Significance:** 1
**Originality:** 3
**Overall Recommendation:** 3
**Confidence:** 5

**Summary:**

This paper introduces LongCoT, a new benchmark designed to evaluate long-horizon chain-of-thought reasoning across five domains: mathematics, chemistry, computer science, logic, and chess. The benchmark consists of compositional and procedural templates that require models to sustain extended reasoning over many substeps.

Experimental results show that even frontier models achieve less than 10% accuracy on the full benchmark, suggesting substantial limitations in current long-horizon reasoning capabilities. The authors further analyze correlations between token usage, DAG size, and performance to study how reasoning degrades as problem complexity increases.

**Compliance With Llm Reviewing Policy:**

Affirmed.

**Final Justification:**

I keep my original score for its limited practical significance, but I respect the AC's decision for acceptance or rejection

**Key Questions For Authors:**

1. Have you considered the impact of individual subproblem difficulty on the overall difficulty of the composed problems?

2. Will you conduct any quantitative error analysis (I noticed some discussion in Appendix C), such as measuring the step at which the first error occurs and analyzing the distribution of corresponding node characteristics?

3. Will you consider introducing long-horizon problems with stronger real-world relevance in the mathematics, chemistry, and computer science domains? For example, a mathematical proof often relies on multiple propositions and lemmas, which naturally form compositional nodes and may carry greater research value.

**Limitations:**

I noticed some discussion in Appendix D.

**Strengths And Weaknesses:**

> (+) denotes the strength, (-) denotes the weakness.

**Soundness：**

（+）The empirical results are relatively comprehensive. The paper reports results on several frontier models and shows that the LongCoT benchmark is quite challenging for them.

（+）Some supplementary analyses are also meaningful, including the correlation between token usage / DAG size and reasoning performance.

（--）A major issue concerns the validity of the claimed correlation between token usage / DAG size and reasoning performance. In the mathematics, chemistry, and computer science domains in particular, subproblems are largely independent, with only answer and condition dependencies linking them. In such cases, the overall reasoning difficulty can be heavily influenced by the intrinsic difficulty of individual subproblems. If a particular subproblem within a composition is especially difficult, it may cause failure regardless of the DAG size. I believe the current conclusions are therefore somewhat misleading.

（-）In addition to the number of DAG nodes, the edge connectivity structure of the DAG also affects complexity. For instance, if the structure is a simple chain, the model can naturally execute it sequentially. I think the paper does not consider this aspect thoroughly enough.

**Presentation：**

（++）The overall writing of the paper is clear, and it is easy to follow.

（-）The name “LongCoT” is not ideal, as it is commonly understood as a paradigm distinguishing it from traditional short CoT. I suggest reconsidering the name.


**Significance：**

（---）I am skeptical about the significance of this benchmark for two reasons:

* As mentioned in the soundness, the control of subproblem difficulty directly affects the validity of the conclusions. If long-horizon reasoning is the primary focus, the difficulty of subproblems should be relatively similar. Otherwise, multiple variables may interfere with each other, undermining the evaluation’s interpretability.

* Among the five domains, the construction of logic and chess problems is natural, with highly coupled substeps. These indeed reflect long-horizon reasoning ability. However, in the mathematics, chemistry, and computer science domains, the problem construction itself lacks strong real-world relevance. Beyond answer and condition dependencies, substeps are otherwise not tightly coupled. From this perspective, these tasks may not fully reflect long-horizon reasoning ability; instead, they resemble single-step reasoning combined with long-context memory management. It is possible to interfere with large models through context memory challenges rather than reasoning itself.


**Originality：**

I do not have major concerns. As long as the benchmark is not a direct reuse of existing datasets, I would not argue against the originality of the data collection.

---

> ### Author Rebuttal · Authors · 2026-03-29
>
> Thank you for your time and valuable feedback! We appreciate the opportunity to address the points raised and further improve our work.
>
> \*__Additional Analysis PDF__: https://shorturl.at/pdhoR
>
> __On subproblem difficulty__. We agree with the reviewer that to isolate some facets of long-horizon reasoning, it is helpful if all subproblems are of similar difficulty. In fact, Figure 5 in the paper demonstrates how accuracy changes with DAG size when we tightly control subproblem difficulty, as all subproblems are drawn from Omni-math, and GPT-5.2 achieves over 98% (pass@1) accuracy on the subset we use (which we uniformly sample from). In this case, when the DAG size reaches 20 nodes (if the difficulty comes from the individual subproblems rather than the long-horizon nature), we should see GPT-5.2 have an accuracy of around 67%, but it is less than half that (see Fig. 1 in the new PDF\*). In general, for both mathematics and chemistry, the pass@1 accuracy over all subquestions is over 95%. For CS, Chess, and Logic, naturally each step is tractable, and it is the size of the problem and its structural complexity that makes solving them challenging (see for e.g. Tables 2 and 3).
>
> It is also true that in genuine long-horizon reasoning, there will be components with varying levels of difficulty, and to evaluate these effects, it seems necessary to include more difficult components in our medium and hard subsets.
>
> __On DAG connectivity structure__. This is an insightful point and we agree that it could strengthen the paper. The current analysis primarily relies on the size of the DAG alone as a proxy for task length/difficulty, when the depth and width are also likely to significantly influence performance. We have performed an analysis of this on the mathematics questions (Figures 3-5 in the new PDF\*), examining accuracy against max-depth crossed with max-width, max-in-degree, and max-out-degree, for both GPT-5.2 and Gemini. The trends are consistent across both models: at lower depths, increased width and degree reduce accuracy substantially, while at higher depths, accuracy is low regardless of structural complexity. We commit to updating the paper with this analysis and also agree that a fine-grained quantitative analysis of initial failure modes would strengthen the paper and plan to include it, though it requires careful human annotation of reasoning traces from open models.
>
> __On the name “LongCoT.”__  We understand the reviewer’s concern. Potential alternative names are “LongCoT-Bench” or “LHRBench”, and we welcome any suggestions.
>
> __On significance and real-world relevance.__ We concur that the logic and chess domains (and most CS questions) provide highly coupled substeps that naturally reflect long-horizon reasoning. Regarding mathematics and chemistry, the problems were designed to rigorously test compositional reasoning, where the solution requires executing an extended sequence of steps, often relying on the output of previous steps (answer and condition dependencies).
>
> To directly test whether the difficulty arises from dependencies or simply from producing long outputs, we run an ablation where the same subproblems are presented within the same DAG structure but with no inter-node dependencies, and score the model on the same leaf nodes (Figure 2 in new PDF\*). On medium-difficulty questions, GPT-5.2 scores 58% on independent questions vs 8% when composed; on hard questions, 55% vs 4%. Token usage is comparable across both conditions. This confirms that compositional dependency structure, not just context length, is the primary driver of failure. We also note that our prompts are ~2k tokens and the extended context is entirely model-generated, so any context-length effects reflect the model's own reasoning process, not an externally imposed memory challenge.
>
> As you note, we do also include more realistic questions, and think that the chemistry questions are reasonably close to an actual synthesis workflow. The explicitly structured questions also enable controlled analysis of properties that might affect long-horizon reasoning, such as DAG width, depth, and degree (see Table 6 and the new PDF\*), which would be impossible with purely natural problems. They are also easier to construct and will allow us to continue updating the benchmark if it gets saturated.
>
> Response to Key Questions:
> 1. See “On subproblem difficulty” above.
> 2. See “On DAG connectivity structure” above.
> 3. While we agree that more tightly-coupled/realistic problems would be useful, they are more time-intensive and costly to develop. We believe that one of the main methodological advances of our paper is demonstrating that interleaving short problems is sufficient to induce long-horizon failures, and will explore more naturalistic problem types in future work.
>
> We sincerely appreciate the reviewer’s feedback and have aimed to address all comments above. We would be grateful if the reviewer would consider raising their score.

---

> > ### Author Rebuttal · Reviewer_qHXU · 2026-04-04
> >
> > Thanks for your response. I appreciate your new experiments on DAG connectivity patterns. However, regarding subproblem difficulty—Omni-math uses a difficulty scale of 1 to 10, and sampling combinations from it would clearly not guarantee consistent difficulty levels. I also did not find any conclusions that directly address my concern. As for the practical significance of the dataset, this is indeed an issue that requires more careful consideration, and at least some discussion should be added to the paper so that the current domains do not appear so disconnected from one another. In summary, I maintain my score and fully respect the AC's decision.

---

> > > ### Author Response · Authors · 2026-04-07
> > >
> > > We appreciate the reviewer’s response. We would like to briefly address the remaining points.
> > >
> > > On subproblem difficulty: the reviewer's concern is that Omni-math's 1-10 scale does not guarantee consistent difficulty levels. But the empirical solve rate is itself the relevant measure of consistency. If GPT-5.2 solves 98% of the individual subproblems, difficulty variance is empirically bounded regardless of what nominal scale they are drawn from. We also note that the reviewer simultaneously flags uncontrolled subproblem difficulty and lack of real-world relevance as weaknesses, but realistic long-horizon problems inherently contain steps of varying difficulty. We include both controlled-difficulty and more realistic settings to address both ends of this spectrum.
> > >
> > > On domain coherence: the reviewer asks that discussion be added so the domains do not appear disconnected, but this discussion is already present in Section 3, which defines four core long-horizon reasoning capabilities and explains how different domains and template types stress-test different subsets.
> > >
> > > Finally, we note that our independence ablation (Figure 2 in the additional analysis) directly addresses the reviewer’s core concern that math and chemistry domains may test context memory management rather than compositional reasoning. Accuracy jumps from 8% to 58% (medium) and 4% to 55% (hard) when dependencies are removed, with comparable token counts. We would be grateful if the reviewer could engage with this result.

---

### Official Review · Reviewer_r96i · 2026-03-11

**Soundness:** 3
**Presentation:** 3
**Significance:** 3
**Originality:** 3
**Overall Recommendation:** 4
**Confidence:** 3

**Summary:**

This paper introduces a new benchmark designed to evaluate the long-horizon ability of large language models (LLMs). The benchmark contains 2,500 expert-designed problems across five domains: mathematics, chemistry, computer science, chess, and logic. Each problem has a relatively short prompt but requires models to produce very long reasoning traces (often tens to hundreds of thousands of tokens) to reach a verifiable final answer. Experimental results show that even the most advanced models perform poorly on the benchmark with GPT5.2 perform 9.8%, revealing a large gap for current LLM capability.

**Compliance With Llm Reviewing Policy:**

Affirmed.

**Key Questions For Authors:**

see weakness

**Limitations:**

yes

**Strengths And Weaknesses:**

Strengths
1. The benchmark is hard and have large improvement room.
2. The benchmark targets at an important and under explored capability and design pipeline to isolate the reasoning ability.
3. The experiment is comprehensive with detailed ablation analysis.

Weaknesses
1. It is unclear whether the extremely long reasoning traces (50k–100k tokens) are necessary to solve these tasks. It would be useful to compare performance with shorter reasoning budgets or alternative prompting strategies to assess whether such long outputs are essential.
2. A large portion of the benchmark tasks are synthetically generated through templates. It is doubtable how well improvements on this benchmark would transfer to real-world reasoning tasks.
3. The evaluation setup solves the entire graph in a single run. It would be interesting to investigate whether alternative evaluation protocols—such as iterative solving, staged prompting, or partial decomposition across multiple runs—could improve performance or better reflect realistic agent workflows.

---

> ### Author Rebuttal · Authors · 2026-03-29
>
> Thank you for your time and valuable feedback! We appreciate the opportunity to address the points raised and further improve our work.
>
> \*__Additional analysis PDF__: https://shorturl.at/pdhoR
>
> __On the necessity of long reasoning traces.__
> This is an important point, and we thank you for raising it. We have now performed an ablation using GPT-5.2 with “low”, “medium” and “high” reasoning efforts on a subset of questions from each domain. We find that accuracy consistently increases with the number of tokens (and reasoning effort) across all domains for this model, providing strong evidence that the length of the traces is necessary for high accuracy. We also try several different prompting strategies with Grok 4.1 Fast (due to our limited API budget) and find very similar or lower accuracies to our baseline prompts, suggesting that this is not a major influence on accuracy or token usage.
>
> _GPT 5.2 Reasoning Effort Ablation on a LongCoT Subset:_
>
> | Domain | Low | Medium | High |
>   |---|---|---|---|
>   | Chess | 2/50 (4K) | 3/50 (15K) | 5/50 (34K) |
>   | Logic | 0/50 (5K) | 1/50 (21K) | 6/50 (79K) |
>   | Chemistry | 0/50 (6K) | 3/50 (17K) | 6/50 (27K) |
>   | Math | 0/50 (12K) | 3/50 (20K) | 6/50 (54K) |
>
>
> _Prompting Strategies for Grok 4.1 Fast Reasoning (on a random subset of 150 problems from LongCoT and LongCoT-mini):_
>
> Current prompt with just the problem and asking the model to reason over it: 13/150\
> Prompt asking the model to reason concisely: 7/150\
> Prompt asking the model to reason step by step and be focused: 13/150\
> Prompt asking the model to act like a subject matter expert when reasoning: 9/150\
> Prompt asking the model to reflect on each important step before proceeding: 12/150\
>
> __On synthetic questions.__
> While some of our problems (particularly mathematics) are constructed from templates, we note two things. First, the model ordering on our benchmark is broadly similar to that on other difficult agentic benchmarks (e.g. terminalbench [1] with the Droid or Terminus 2 agents or Tau-bench [2]), suggesting that the skills we directly test and isolate here do transfer to agentic settings. Second, the controlled structure is a design choice since it enables us to test a very wide range of reasoning structures across several domains and problem types (see Figure 2). It also allows us to systematically analyse how properties such as DAG depth, width, and dependency type affect performance (see Table 6 in the original paper and Figures 3-5 in our new analysis PDF attached*), which would be challenging to do with purely natural problems. We also note that many questions are not synthetic, for e.g. the chemistry questions mirror potential real-world synthesis workflows and are developed by experts in the field.
>
> __On iterative solving.__ This is another excellent point, and we believe that our RLM (Recursive Language Models) [3] ablations in Section 4.2 address this. Using an RLM (without code execution for sub-agents), models are allowed to decompose the question (context) and have sub-questions answered via iterative staged LLM subcalls to avoid filling up their own context. However, we find that this leads to little or no improvement, suggesting the models are not able to decompose the questions correctly and plan out their reasoning. This also suggests that the difficulty of the benchmark is not just related to managing long inputs but also stress tests compositional reasoning and other skills.
>
> With code-use enabled for RLM sub-agents, the model solves some more logic and chess questions (as it can simply write brute force solvers), but still struggles a lot on maths, chemistry, and computer science, again suggesting that correct decomposition is difficult even for frontier models, and more capabilities are necessary for long-horizon reasoning.
>
> [1] Terminal-Bench: Benchmarking Agents on Hard, Realistic Tasks in Command Line Interfaces, Merrill et. al.\
> [2] T-bench: A Benchmark for Tool-Agent-User Interaction in Real-World Domains, Yao et. al.\
> [3] Recursive Language Models, Zhang et. al.\
>
> We sincerely appreciate the reviewer’s feedback and have aimed to address all the comments above. We would be very grateful if the reviewer would consider raising their score.

---

> > ### Author Rebuttal · Reviewer_r96i · 2026-04-02
> >
> > Thank you

---

> > > ### Author Response · Authors · 2026-04-06
> > >
> > > Dear Reviewer,
> > >
> > > Thank you for your constructive feedback, which has meaningfully strengthened the paper. We conducted a rigorous set of new experiments to thoroughly address each of your points, including a reasoning effort ablation across all domains, a systematic comparison of prompting strategies, and a detailed discussion of iterative solving through our RLM results. These experiments strongly support our original findings and hypotheses. We would be very grateful if you would consider raising your score in support of our work and are happy to conduct any additional experiments.

---

### Official Review · Reviewer_M24t · 2026-03-13

**Soundness:** 3
**Presentation:** 4
**Significance:** 3
**Originality:** 3
**Overall Recommendation:** 5
**Confidence:** 3

**Summary:**

This paper introduces LongCoT, a benchmark for evaluating frontier LLMs on long-horizon reasoning tasks. The dataset includes 2500 questions across 5 categories. Evaluation of frontier models shows that the benchmark poses a significant challenge.

**Compliance With Llm Reviewing Policy:**

Affirmed.

**Final Justification:**

I will keep my positive assessment and recommend acceptance.

**Key Questions For Authors:**

See weaknesses.

**Limitations:**

Yes

**Strengths And Weaknesses:**

Summary: This paper introduces LongCoT, a benchmark for evaluating frontier LLMs on long-horizon reasoning tasks. The dataset includes 2500 questions across 5 categories. Evaluation of frontier models shows that the benchmark poses a significant challenge.

Strengths:
1. The template-based problem construction is neat. It offers a clear way to analyze and scale the difficulty of the problems; the latter is especially important when the current version becomes saturated.
2. LongCoT poses a significant challenge to current frontier models, with state-of-the-art models scoring <10%. Similar challenging benchmarks (e.g., HLE) entangle reasoning with domain knowledge and tool use, while LongCoT is largely resistant.
3. The reasoning analysis of Kimi-K2 and DeepSeek-V3.2 is interesting, but I would have appreciated more discussion on how this analysis was conducted.

Weaknesses:
1. Some technical details are unclear. For instance, how is each local step ensured to be tractable? Similarly, it's not described how the decontamination on the math subset was done.
2. It would be valuable to see more frontier models evaluated, but this is understandably difficult given the high evaluation cost.
3. Claims like "models that use more of their available reasoning budget do markedly better" are undersupported; for instance, Kimi-K2 and GPT-5.2 both reason for ~60k tokens per task, but differ greatly in performance (1.86% vs 9.83%). It would be more valuable to see GPT-5.2's performance at different reasoning levels (e.g., low, medium, high, etc.).

---

> ### Author Rebuttal · Authors · 2026-03-29
>
> Thank you for your time and valuable feedback! We appreciate the opportunity to address the points raised and further improve our work.
>
> __On tractable subproblems.__
> For questions that are made up of explicit subquestions, we evaluate the pass@1 accuracy of GPT-5.2 on the individual subproblems themselves to ensure that accuracy is high. For both mathematics and chemistry, the pass@1 accuracy over all subquestions is over 95%. We also study the case where the mathematics questions are made up of only a subset of Omni-MATH questions (where we validate pass@1 accuracy of GPT-5.2 is 98%), and find that the long-horizon accuracy of chained problems is significantly lower than what would be achieved if the errors were independent (30% vs 67% at n=20, see Figure 5). For CS, Chess, and Logic, each step is naturally tractable, and it is the size of the problem and the complexity of its structure that makes solving them challenging (see, for example, Tables 2 and 3). This allows us to ensure that most failures reflect long-horizon reasoning limitations.
>
> For the math subquestions, we perform an LLM judge-based question incompleteness check, question validity check, filter questions with images, and test a partial prompt strategy as a simple check for memorisation, similar to [1]. Beyond this, it is hard to guarantee that some of the questions are not seen in the training corpus, but we source only math questions from recent high-quality benchmarks, and models perform poorly on our long-horizon reasoning problems.
>
> __On additional models.__
> We agree that it would be useful to see more (and more recent) frontier models evaluated on the benchmark, and we have done our best to evaluate a large set for this paper. Our goal is to publicly release our work and encourage more models to be evaluated, given the ease of using our benchmark. We will also release the code to make running the benchmark as easy as possible, to allow providers to submit their own results.
>
> __On the number of tokens and reasoning effort.__
> This is a very good point, and we thank the reviewer for raising it. At your suggestion, we performed an ablation over a subset of questions from our domains with GPT-5.2, across reasoning levels “low”, “medium”, and “high”. We find that a lower reasoning level uniformly results in less token usage and lower accuracy, providing better evidence for the claim. We agree that this relationship holds primarily with respect to a given model. Cross-model differences at similar token counts (e.g. Kimi-K2 vs GPT-5.2) likely reflect differences in reasoning quality per token, which makes the benchmark useful for discriminating reasoning capability, not just verbosity. We will clarify this and include this study in our camera-ready version.
>
> | Domain | Low | Medium | High |
>   |---|---|---|---|
>   | Chess | 2/50 (4K) | 3/50 (15K) | 5/50 (34K) |
>   | Logic | 0/50 (5K) | 1/50 (21K) | 6/50 (79K) |
>   | Chemistry | 0/50 (6K) | 3/50 (17K) | 6/50 (27K) |
>   | Math | 0/50 (12K) | 3/50 (20K) | 6/50 (54K) |
>
> We sincerely appreciate the reviewer’s feedback and have aimed to address all the comments above.
>
> [1] Reasoning or Memorization? Unreliable Results of Reinforcement Learning Due to Data Contamination, Wu et. al.

---

> > ### Author Rebuttal · Reviewer_M24t · 2026-04-04
> >
> > The authors have addressed by concerns. I will keep my positive assessment.

---

### Decision · Program_Chairs · 2026-04-30

**Decision:**

Accept (regular)

**Comment:**

The paper contributes a benchmark for an important problem: how do LLMs perform in reasoning over long CoT horizons? While long input context is widely studied, the paper makes an effort to benchmark reasoning over long output context length. Reviewers appreciated the template-based construction, and the fact that the benchmark is far from saturated--even frontier models achieve lower than 10%. The experiments are extensive covering many different models.

The authors rebuttal clarified some concerns, especially on difficulty of individual problems versus the full composite problem. I recommend that the authors include this ablation in the final version. Overall, I think the paper makes a useful contribution to understanding the reasoning limits of LLMs and improving it.